# Responsible Disclosure of Generative Models Using Scalable Fingerprinting

**Ning Yu**[1,2,3†*]   **Vladislav Skripniuk**[4†]   **Dingfan Chen**[4]   **Larry Davis**[2]   **Mario Fritz**[4]
[1]Salesforce Research   [2]University of Maryland   [3]Max Planck Institute for Informatics
[4]CISPA Helmholtz Center for Information Security
`ning.yu@salesforce.com  vladislav@mpi-inf.mpg.de`
`{dingfan.chen, fritz}@cispa.de  lsd@cs.umd.edu`

## Abstract

Over the past years, deep generative models have achieved a new level of performance. Generated data has become difficult, if not impossible, to be distinguished from real data. While there are plenty of use cases that benefit from this technology, there are also strong concerns on how this new technology can be misused to generate deep fakes and enable misinformation at scale. Unfortunately, current deep fake detection methods are not sustainable, as the gap between real and fake continues to close. In contrast, our work enables a responsible disclosure of such state-of-the-art generative models, that allows model inventors to fingerprint their models, so that the generated samples containing a fingerprint can be accurately detected and attributed to a source. Our technique achieves this by an efficient and scalable ad-hoc generation of a large population of models with distinct fingerprints. Our recommended operation point uses a 128-bit fingerprint which in principle results in more than $10^{38}$ identifiable models. Experiments show that our method fulfills key properties of a fingerprinting mechanism and achieves effectiveness in deep fake detection and attribution. Code and models are available at GitHub.

## 1 Introduction

Over the recent seven years, deep generative models have demonstrated stunning performance in generating photorealistic images, considerably boosted by the revolutionary technique of generative adversarial networks (GANs) (Goodfellow et al., 2014; Radford et al., 2016; Gulrajani et al., 2017; Miyato et al., 2018; Brock et al., 2018; Karras et al., 2018; 2019; 2020).

However, with the closing margins between real and fake, a flood of strong concerns arise (Harris, 2018; Chesney & Citron, 2019; Brundage et al., 2018): how if these models are misused to spoof sensors, generate deep fakes, and enable misinformation at scale? Not only human beings have difficulties in distinguishing deep fakes, but dedicated research efforts on deep fake detection (Durall et al., 2019; Zhang et al., 2019; Frank et al., 2020; Zhang et al., 2020) and attribution (Marra et al., 2019; Yu et al., 2019; Wang et al., 2020) are also unable to sustain longer against the evolution of generative models. For example, researchers delve into details on how deep fake detection works, and learn to improve generation that better fits the detection criteria (Zhang et al., 2020; Durall et al., 2020). In principle, any successful detector can play an auxiliary role in augmenting the discriminator in the next iteration of GAN techniques, and consequently results in an even stronger generator.

The dark side of deep generative models delays its industrialization process. For example, when commercializing the GPT-2 (Radford et al., 2019) and GPT-3 (Brown et al., 2020) models, OpenAI leans conservative to open-source their models but rather only release the black-box APIs[1]. They involve expensive human labor in the loop to review user downloads and monitor the usage of their APIs. Yet still, it is a challenging and industry-wide task on how to trace the responsibility of the downstream use cases in an open end.

To pioneer in this task, we propose a model fingerprinting mechanism that enables responsible release and regulation of generative models. In particular, we allow responsible model inventors to

---

[*]This work was done when Ning Yu was in a joint Ph.D. program with the University of Maryland and Max Planck Institute for Informatics.

[†]Equal contribution.

[1]https://openai.com/blog/openai-api/

fingerprint their generators and disclose their responsibilities. As a result, the generated samples contain fingerprints that can be accurately detected and attributed to their sources. This is achieved by an efficient and scalable ad-hoc generation of a large population of generator instances with distinct fingerprints. See Figure 1 Middle.

Similar in the spirit of the dynamic filter networks (Jia et al., 2016) and style-based generator architectures (Karras et al., 2019; 2020) where their network filters are not freely learned but conditioned on an input, we learn to parameterize a unique fingerprint into the filters of each generator instance. The core gist is to incorporate a fingerprint auto-encoder into a GAN framework while preserving the original generation performance. See Figure 1 Left. In particular, given

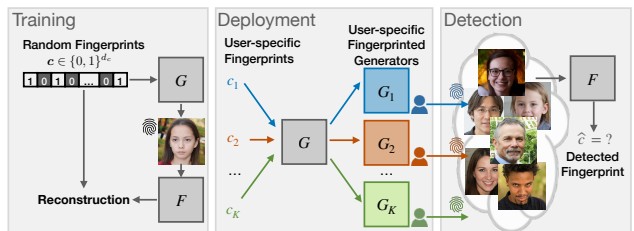

Figure 1: The diagram of our fingerprinting mechanism for generators. See main text for descriptions.

a GAN backbone, we use the fingerprint embedding from the encoder to modulate each convolutional filter of the generator (Figure 2(b)), and try to decode this fingerprint from the generated images. We jointly train the fingerprint auto-encoder and GAN with our fingerprint related losses and the original adversarial loss. See Figure 2(a) for the diagram, and 3 for details.

After training, the responsible model inventor is capable of efficiently fingerprinting and releasing different generator instances to different user downloads, which are equipped with the same generation performance but with different fingerprints. Each user download corresponds to a unique fingerprint, which is maintained by the inventor's database. As a result, when misuse of a model happens, the model inventor can use the decoder to detect the fingerprint from the generated images, match it in the database, and then trace the responsibility of the user. See Figure 1 Right. Based on this form of responsible disclosure, responsible model inventors, like OpenAI, have a way to mitigate adverse side effects on society when releasing their powerful models, while at the same time should have an automatic way to attribute misuses.

There are several key properties of our mechanism. The **efficiency** to instantiate a generator is inherently satisfied because, after training, the fingerprint encoding and filter modulation run with little overhead. We evaluate the **effectiveness** of our fingerprinting and obtain almost perfect fingerprint detection accuracy. We also justify the **fidelity** with a negligible side effect on the original generation quality. See Section 4.1. Our recommended operation point uses a 128-bit fingerprint (Section 4.2) which in principle results in a **capacity** of more than $10^{38}$ identifiable generator instances. The **scalability** benefits from the fact that fingerprints are randomly sampled on the fly during training so that fingerprint detection generalizes well for the entire fingerprint space. See Section 4.3 for validation. In addition, we validate in Section 4.4 the **secrecy** of presence and value of our fingerprints against shadow model attacks. We validate in Section 4.5 the **robustness** and **immunizability** against perturbation on generated images.

To target the initial motivation, we move the deep fake detection and attribution solutions from *passive* detectors to *proactive* fingerprinting. We show in Section 4.6 saturated performance and advantages over two state-of-the-art discriminative methods (Yu et al., 2019; Wang et al., 2020) especially in the open world. This is because, conditioned on user-specific fingerprint inputs, the presence of such fingerprints in generated images guarantees the margin between real and fake, and facilitates the attribution and responsibility tracing of deep fakes to their sources.

Our **contributions** are in four thrusts: (1) We enhance the concept of fingerprinting for generative models that enables a responsible disclosure of state-of-the-art GAN models. (2) We pioneer in the novel direction for efficient and scalable GAN fingerprinting mechanism, i.e., only one generic GAN model is trained while more than $10^{38}$ fingerprinted generator instances can be obtained with little overhead during deployment. (3) We also justify several key properties of our fingerprinting, including effectiveness, fidelity, large capacity, scalability, secrecy, robustness, and immunizability. (4) Finally, for the deep fake detection and attribution tasks, we move the solution from *passive* classifiers to *proactive* fingerprinting, and validate its saturated performance and advantages. It makes our responsible disclosure independent of the evolution of GAN techniques.

## 2 RELATED WORK

**Deep fake detection and attribution.** These tasks come along with the increasing concerns on deep fake misuse (Harris, 2018; Chesney & Citron, 2019; Brundage et al., 2018). Deep fake detection is a binary classification problem to distinguish fake samples from real ones, while attribution further traces their sources. The findings of visually imperceptible but machine-distinguishable patterns in GAN-generated images make these tasks viable by noise pattern matching (Marra et al., 2019), deep classifiers (Afchar et al., 2018; Hsu et al., 2018; Yu et al., 2019), or deep Recurrent Neural Networks (Güera & Delp, 2018). (Zhang et al., 2019; Durall et al., 2019; 2020; Liu et al., 2020) observe that mismatches between real and fake in frequency domain or in texture representation can facilitate deep fake detection. (Wang et al., 2020; Girish et al., 2021) follow up with generalization across different GAN techniques towards open world. Beyond attribution, (Albright & McCloskey, 2019; Asnani et al., 2021) even reverse the engineering to predict in the hyper-parameter space of potential generator sources.

However, these *passive* detection methods heavily rely on the inherent clues in deep fakes. Therefore, they can barely sustain a long time against the adversarial iterations of GAN techniques. For example, (Durall et al., 2020) improves generation realism by closing the gap in generated high-frequency components. To handle this situation, artificial fingerprinting is proposed in (Yu et al., 2021) to *proactively* embed clues into generative models by rooting fingerprints into training data. This makes deep fake detection independent of GAN evolution. Yet, as *indirect* fingerprinting, (Yu et al., 2021) cannot scale up to a large number of fingerprints because they have to pre-process training data for each individual fingerprint and re-train a generator with each fingerprint. Our method is similar in spirit to (Yu et al., 2021), but possesses fundamental advantages by *directly* and *efficiently* fingerprinting generative models: after training one generic fingerprinting model, we can instantiate a large number of generators ad-hoc with different fingerprints.

**Image steganography and watermarking.** Steganography targets to manipulate carrier images in a hidden manner such that the communication through the images can only be understood by the sender and the intended recipient (Fridrich, 2009). Traditional methods rely on Fourier transform (Cox et al., 2002; Cayre et al., 2005), JPEG compression[2][3], or least significant bits modification (Pevnỳ et al., 2010; Holub et al., 2014). Recent works utilize deep neural encoder and decoder to hide information (Baluja, 2017; Tancik et al., 2020; Luo et al., 2020). Watermarking targets to embed ownership information into carrier images such that the owner's identity and authenticity can be verified. It belongs to a form of steganography that sometimes interacts with physical images (Tancik et al., 2020). Existing methods rely on log-polar frequency domain (Pereira & Pun, 2000; Kang et al., 2010), printer-camera transform (Solanki et al., 2006; Pramila et al., 2018), or display-camera transform (Yuan et al., 2013; Fang et al., 2018). Recent works also use deep neural networks to detect when an image has been re-imaged (Fan et al., 2018; Tancik et al., 2020). The concept and function of our fingerprinting solution is similar in spirit of watermarking, but differs fundamentally. In particular, we did not retouch individual images. Rather, our solution is the first to retouch generator parameters so as to encode information into the model.

**Network watermarking.** Network watermarking techniques (Uchida et al., 2017; Adi et al., 2018; Zhang et al., 2018; Chen et al., 2019; Rouhani et al., 2019; Ong et al., 2021; Yu et al., 2021) embed watermarks into network parameters rather than pixels while not deteriorating the original utility. Our solution shares motivations with them but substantially differs in terms of concepts, motivations, and techniques. For concepts, most existing works are applicable to only image classification models, only (Ong et al., 2021; Yu et al., 2021) work for generative models but suffer from poor efficiency and scalability. For motivations, the existing works target to fingerprint a single model, while we are motivated by the limitation of (Ong et al., 2021; Yu et al., 2021) to scale up the fingerprinting to as many as $10^{38}$ various generator instances within one-time training. For techniques, most existing works embed fingerprints in the input-output behaviors (Adi et al., 2018; Zhang et al., 2018; Ong et al., 2021), while our solution gets rid of such trigger input for improved scalability.

---

[2] http://www.outguess.org/
[3] http://steghide.sourceforge.net

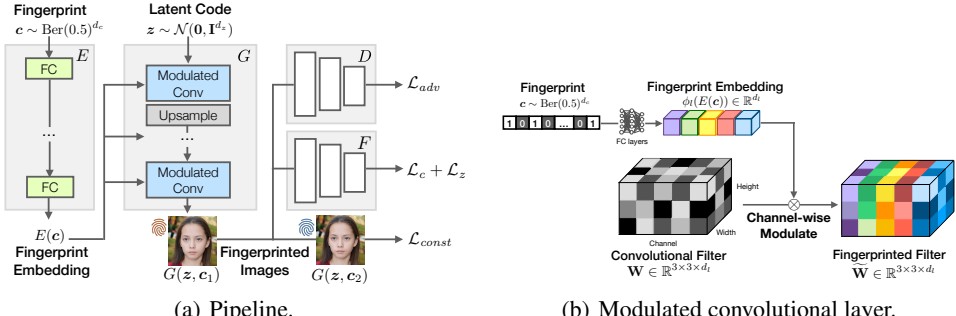

(a) Pipeline.

(b) Modulated convolutional layer.

Figure 2: The diagrams of our fingerprinting pipeline and the modulated convolutional layer.

## 3 GAN FINGERPRINTING NETWORKS

Throughout the paper, we stand for the responsible model inventor's perspective, which is regarded as the regulation hub of our experiments. None of the encoder, decoder, and training data should be accessible to the public. Only fingerprinted generator instances are released to the open end.

We list symbol notations at the beginning. We use latent code $z \sim \mathcal{N}(\mathbf{0}, \mathbf{I}^{d_z})$ to control generated contents. We set $d_z = 512$. We represent fingerprint $c \sim \text{Ber}(0.5)^{d_c}$ as a sequence of bits. It follows Bernoulli distribution with probability $0.5$. We non-trivially choose the fingerprint length $d_c$ in Section 4.2. We denote encoder $E$ mapping $c$ to its embedding, generator $G$ mapping $(z, E(c))$ to the image domain, discriminator $D$ mapping an image $x \sim p_{\text{data}}$ to the real/fake classification probability, and decoder $F$ mapping an image to the decoded latent code and fingerprint $(\widehat{z}, \widehat{c})$. In the following formulations, we denote $G(z, E(c))$ as $G(z, c)$ for brevity.

### 3.1 PIPELINE

We consider three goals in our training. First, we preserve the original functionality of GANs to generate realistic images, as close to real distribution as possible. We use the unsaturated logistic loss as in (Goodfellow et al., 2014; Karras et al., 2019; 2020) for real/fake binary classification:

$$\mathcal{L}_{adv} = \mathop{\mathbb{E}}_{x \sim p_{\text{data}}} \log D(x) + \mathop{\mathbb{E}}_{\substack{z \sim \mathcal{N}(\mathbf{0}, \mathbf{I}^{d_z}) \\ c \sim \text{Ber}(0.5)^{d_c}}} \log \Big( 1 - D\big(G(z, c)\big) \Big) \tag{1}$$

In addition, similar to (Srivastava et al., 2017), we reconstruct the latent code through the decoder $F$ to augment generation diversity and mitigate the mode collapse issue of GANs (Srivastava et al., 2017; Li & Malik, 2018; Yu et al., 2020).

$$\mathcal{L}_z = \mathop{\mathbb{E}}_{\substack{z \sim \mathcal{N}(\mathbf{0}, \mathbf{I}^{d_z}) \\ c \sim \text{Ber}(0.5)^{d_c}}} \sum_{k=1}^{d_z} \Big( z_k - F\big(G(z, c)\big)_k \Big)^2 \tag{2}$$

where we use the first $d_z$ output elements of $F$ that correspond to the decoded latent code.

The second goal is to reconstruct the fingerprint so as to allow fingerprint detection.

$$\mathcal{L}_c = \mathop{\mathbb{E}}_{\substack{z \sim \mathcal{N}(\mathbf{0}, \mathbf{I}^{d_z}) \\ c \sim \text{Ber}(0.5)^{d_c}}} \sum_{k=1}^{d_c} \Big[ c_k \log \sigma\Big( F\big(G(z, c)\big)_{d_z+k} \Big) + (1 - c_k) \log \Big( 1 - \sigma\Big( F\big(G(z, c)\big)_{d_z+k} \Big) \Big) \Big] \tag{3}$$

where we use the last $d_c$ output elements of $F$ as the decoded fingerprint. $\sigma(\cdot)$ denotes the sigmoid function that differentiably clips the output to the range of $[0, 1]$. The reconstruction is therefore a combination of cross-entropy binary classification for each bit.

It is worth noting that we use one decoder to decode both the latent code and fingerprint, which benefits explicit **disentanglement** between their representations as discussed below.

The third goal is to disentangle the representation between latent code and fingerprint. Desirably, latent code should have exclusive control over the generated content. This sticks to the original

generation functionality. Therefore, two images with different fingerprints but with identical latent code should have a consistent appearance. We formulate the consistency loss as:

$$\mathcal{L}_{const} = \mathop{\mathbb{E}}_{\substack{\boldsymbol{z} \sim \mathcal{N}(\boldsymbol{0}, \mathbf{I}^{d_z}) \\ \boldsymbol{c}_1, \boldsymbol{c}_2 \sim \text{Ber}(0.5)^{d_c}}} \|G(\boldsymbol{z}, \boldsymbol{c}_1) - G(\boldsymbol{z}, \boldsymbol{c}_2)\|_2^2 \tag{4}$$

The disentangled effect is demonstrated in Figure 3 and Appendix Figure 6.

Our final training objective is as follows. We optimize it under the adversarial training framework w.r.t. $E$, $G$, $F$, and $D$.

$$\min_{E,F,G} \max_{D} \lambda_1 \mathcal{L}_{adv} + \lambda_2 \mathcal{L}_z + \lambda_3 \mathcal{L}_c + \lambda_4 \mathcal{L}_{const} \tag{5}$$

where $\lambda_1 = 1.0$, $\lambda_2 = 1.0$, $\lambda_3 = 2.0$, and $\lambda_4 = 2.0$ are hyper-parameters to balance the magnitude of each loss term, Each loss contributes to a property of our solution. The weight settings are empirically not sensitive within its magnitude level. See Figure 2(a) for the diagram.

### 3.2 FINGERPRINT MODULATION

At the architectural level, it is non-trivial how to embed $E(\boldsymbol{c})$ into $G$. The gist is to embed fingerprints into the generator parameters rather than generator input, so that after training a generic model we can instantiate a large population of generators with different fingerprints. This is critical to make our fingerprinting efficient and scalable, as validated in Section 4.3. We then deploy only the fingerprinted generator instances to user downloads, not including the encoder.

We achieve this by modulating convolutional filters in the generator backbone with our fingerprint embedding, similar in spirit of (Karras et al., 2020). Given a convolutional kernel $\mathbf{W} \in \mathbb{R}^{3 \times 3 \times d_l}$ at layer $l$, we first project the fingerprint embedding $E(\boldsymbol{c})$ through an affine transformation $\phi_l$ such that $\phi_l(E(\boldsymbol{c})) \in \mathbb{R}^{d_l}$. The transformation is implemented as a fully-connect neural layer with learnable parameters. We then scale each channel of $\mathbf{W}$ with the corresponding value in $\phi_l$. In specific,

$$\widetilde{\mathbf{W}}_{i,j,k} = \phi_l\big(E(\boldsymbol{c})\big)_k \cdot \mathbf{W}_{i,j,k}, \quad \forall i, j, k \tag{6}$$

See Figure 2(b) for a diagram illustration. We compare to the other fingerprint embedding architectures in Section 4.1 and validate the advantages of this one. We conduct modulation for all the convolutional filters at layer $l$ with the same fingerprint embedding. And we investigate in Appendix Section A.2 at which layer to modulate we can achieve the optimal performance. A desirable trade-off is to modulate all convolutional layers.

Note that, during training, latent code $\boldsymbol{z}$ and fingerprint $\boldsymbol{c}$ are jointly sampled. Yet for deployment, the model inventor first samples a fingerprint $\boldsymbol{c}_0$, then modulates the generator $G$ with $\boldsymbol{c}_0$, and then deploys only the modulated generator $G(\cdot, \boldsymbol{c}_0)$ to a user. For that user there allows only one input, i.e. the latent code, to the modulated generator. Once a misuse happens, the inventor uses the decoder to decode the fingerprint and attribute it to the user, so as to achieve responsible disclosure.

## 4 EXPERIMENTS

**Datasets**. We conduct experiments on CelebA face dataset (Liu et al., 2015), LSUN Bedroom and Cat datasets (Yu et al., 2015). LSUN Cat is the most challenging one reported in StyleGAN2 (Karras et al., 2020). We train/evaluate on 30k/30k CelebA, 30k/30k LSUN Bedroom at the size of $128 \times 128$, and 50k/50k LSUN Cat at the size of $256 \times 256$.

**GAN backbone**. We build upon the most recent state-of-the-art StyleGAN2 (Karras et al., 2020) config E. This aligns to the settings in (Yu et al., 2021) and facilitates our direct comparisons. See Appendix for the implementation details.

### 4.1 EFFECTIVENESS AND FIDELITY

**Evaluation.** The effectiveness indicates that the input fingerprints consistently appear in the generated images and can be accurately detected by the decoder. This is measured by fingerprint detection bitwise accuracy over 30k random samples (with random latent codes and random fingerprint codes). We use 128 bits to represent a fingerprint. This is a non-trivial setting as analyzed in Section 4.2.

In addition, bit matching may happen by chance. Following (Yu et al., 2021), we perform a null hypothesis test to evaluate the chance, the lower the more desirable. Given the number of matching bits $k$ between the decoded fingerprint and its encoded ground truth, the null hypothesis $H_0$ is getting

| Method | CelebA | | | LSUN Bedroom | | | LSUN Cat | | |
|---|---|---|---|---|---|---|---|---|---|
| | Bit acc ⇑ | $p$-value ⇓ | FID ⇓ | Bit acc ⇑ | $p$-value ⇓ | FID ⇓ | Bit acc ⇑ | $p$-value ⇓ | FID ⇓ |
| StyleGAN2 | - | - | 9.37 | - | - | 19.24 | - | - | 31.01 |
| outguess | 0.533 | 0.268 | 10.02 | 0.526 | 0.329 | 20.15 | 0.523 | 0.329 | 32.30 |
| steghide | 0.535 | 0.268 | 9.48 | 0.530 | 0.268 | 19.77 | 0.541 | 0.213 | 31.67 |
| (Yu et al., 2021) | 0.989 | $< 10^{-36}$ | 14.13 | 0.983 | $< 10^{-34}$ | 21.31 | 0.990 | $< 10^{-36}$ | 32.60 |
| Ours | 0.991 | $< 10^{-36}$ | 11.50 | 0.993 | $< 10^{-36}$ | 20.50 | 0.996 | $< 10^{-36}$ | 33.94 |
| Ours Variant I | 0.999 | $< 10^{-38}$ | 12.98 | 0.999 | $< 10^{-38}$ | 20.68 | 0.500 | 0.535 | 34.23 |
| Ours Variant II | 0.987 | $< 10^{-36}$ | 13.86 | 0.927 | $< 10^{-25}$ | 21.70 | 0.869 | $< 10^{-17}$ | 34.33 |
| Ours Variant III | 0.990 | $< 10^{-36}$ | 22.59 | 0.896 | $< 10^{-21}$ | 64.91 | 0.901 | $< 10^{-23}$ | 51.74 |

Table 1: Fingerprint detection in bitwise accuracy with $p$-value to accept the null hypothesis test, and generation fidelity in FID. ⇑/⇓ indicates a higher/lower value is more desirable. The baseline results are directly copied from (Yu et al., 2021).

this number of matching bits by chance. It is calculated as $Pr(X > k|H_0) = \sum_{i=k}^{d_c} \binom{d_c}{i} 0.5^{d_c}$, according to the binomial probability distribution with $d_c$ trials, where $d_c$ is the fingerprint bit length. $p$-value should be lower than 0.05 to reject the null hypothesis.

The fidelity reflects how imperceptibly the original generation is affected by fingerprinting. It also helps avoid one's suspect of the presence of fingerprints which may attract adversarial fingerprint removal. We report Fréchet Inception Distance (FID) (Heusel et al., 2017) between 30k generated images and 30k real testing images. A lower value indicates the generated images are more realistic.

**Baselines.** We compare seven baseline methods. The first baseline is the StyleGAN2 (Karras et al., 2020) backbone. It provides the upper bound of fidelity while has no fingerprinting functionality.

The second baseline is (Yu et al., 2021) which is the other proactive but *indirect* fingerprinting method for GANs. Another two baselines, outguess[4] and steghide[5], are similar to (Yu et al., 2021). They just replace the deep image fingerprinting auto-encoder in (Yu et al., 2021) with traditional JPEG-compression-based image watermarking techniques, and still suffer from low efficiency/scalability.

We also compare our mechanism to three architectural variants. The motivation of these variants is to incorporate fingerprints in different manners. Variant I: modulating convolutional filters with only latent code embedding, while instead feeding the fingerprint code through the input of the generator. This is to test the necessity of fingerprint modulation. Variant II: modulating filters twice, with latent code embedding and fingerprint code embedding separately. Variant III: modulating filters with the embedding from the concatenation of latent code and fingerprint code.

**Results.** From Table 1, we find that:

(1) The two traditional image watermarking methods, outguess and steghide, fail to deliver fingerprints in generated images, indicated by the random guess ($\sim 0.5$) detection accuracy. We attribute this to the representation gap between deep generative models and shallow watermarking techniques.

(2) On CelebA, all the other methods achieve almost perfect fingerprint detection accuracy with $p$-value close to zero. This is because CelebA is a landmark-aligned dataset with limited diversity. Fingerprinting synergizes well with generation regardless of model configuration.

(3) On LSUN Bedroom and Cat, only (Yu et al., 2021) and our optimal model obtain saturated fingerprint detection accuracy. Ours Variant I, II, and III do not always achieve saturated performance. Especially Ours Variant I fails on LSUN Cat. We reason that filter modulation is a strong formulation for reconstruction. Modulating fingerprints is necessary for their detection while modulating latent code along with fingerprint code distracts fingerprint reconstruction.

(4) Our method has comparable performance to (Yu et al., 2021), plus substantial advantages in practice: during deployment, we can fingerprint a generator instance in **5 seconds**, in contrast to (Yu et al., 2021) that has to retrain a generator instance in **3-5 days**. This is a **50000×** gain of efficiency.

(5) Our method results in negligible ≤2.93 FID degradation. This is a worthy trade-off to introduce the fingerprinting function.

---

[4] http://www.outguess.org/
[5] http://steghide.sourceforge.net

(6) We show in Figure 3 and Appendix Figure 6 uncurated generated samples from several generator instances. Image qualities are high. Fingerprints are imperceptible. Thanks to the consistency loss $\mathcal{L}_{const}$ in Eq. 4, different generator instances can generate identical images given the same latent code. Their fingerprints are clued only in the non-salient background and distinguishable by our decoder.

## 4.2 CAPACITY

The capacity indicates the number of unique fingerprints our mechanism can accommodate without crosstalk between two fingerprints. This is determined by $d_c$, fingerprint bit length, and by our detection accuracy (according to Section 4.1). The choice of fingerprint bit length is however non-trivial. A longer length can accommodate more fingerprints but is more challenging to reconstruct/detect.

To figure out the optimal fingerprint bit length, we conduct the following experiments. On one hand, given one length, we evaluate our detection accuracy. On the other hand, we estimate the bottom-line requirement for detection accuracy. This is simulated as the maximal percentage of bit overlap among a large bag (1 million) of fingerprint samples. The gap between the detection accuracy and bottom-line requirement should be the larger the better.

In Figure 4, we vary the fingerprint bit length in the options of $\{32, 64, 128, 256, 512\}$, and plot the bitwise detection accuracy in red and the bottom line requirement in blue. We find:

(1) The bottom line requirement is monotonically decreasing w.r.t. the bit length of fingerprint because a larger bit length leads to less heavy fingerprint overlaps.

(2) The testing accuracy is also monotonically decreasing w.r.t. the bit length of fingerprints. This is due to the challenge of fingerprint reconstruction/detection.

(3) The testing accuracy is empirically decreasing more slowly at the beginning and then faster than its bottom-line requirement. We, therefore, pick the bit length 128 as the optimal choice for the maximal gap. We stick to this for all our experiments.

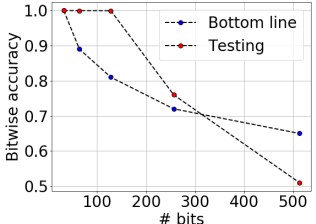

Figure 3: Fidelity and disentangled control on CelebA: generated samples from five generator instances. For each row, we use a unique fingerprint to instantiate a generator. For each column, we feed in the same latent code to the generator instances. More samples are in Appendix Figure 6.

Figure 4: Capacity: fingerprint detection bitwise accuracy and its bottom line requirement w.r.t. fingerprint bit length on CelebA.

(4) Considering our detection bitwise accuracy $\geq 0.991$ and our fingerprint bit length as 128, we derive in principle our mechanism can hold a large capacity of $2^{128 \times 0.991} \approx 10^{38}$ identifiable fingerprints.

## 4.3 SCALABILITY

Scalability is one of the advantageous properties of our mechanism: during training, we can efficiently instantiate a large capacity of generators with arbitrary fingerprints on the fly, so that fingerprint detection generalizes well during testing. To validate this, we compare to the baselines where we intentionally downgrade our method with access to only a finite set of fingerprints. These baselines stand for the category of non-scalable fingerprinting methods that have to re-train a generator instance for each fingerprint, e.g. (Yu et al., 2021). We cannot directly compare to (Yu et al., 2021) because it is impractical (time-consuming) to instantiate a large number of their generators for analysis. As a workaround, we trained our detector using $\leq$1k real samples to simulate the non-scalable nature of the baseline.

| Fingerprint set size | Training acc $\Uparrow$ | Testing acc $\Uparrow$ |
|---|---|---|
| 10 | 1.000 | 0.512 |
| 100 | 1.000 | 0.537 |
| 1k | 1.000 | 0.752 |
| 10k | 0.990 | 0.988 |
| 100k | 0.983 | 0.981 |
| Sampling on the fly | 0.991 | 0.991 |

Table 2: Scalability: fingerprint detection in bitwise accuracy w.r.t. set size on CelebA. $\Uparrow$ indicates a higher value is more desirable.

From Table 2 we show that fingerprint detection fails to generalize unless we can instantiate generators with 10k or more fingerprint samples. This indicates the necessity to equip GANs with an efficient and scalable fingerprinting mechanism, preferably the one on the fly.

### 4.4 SECRECY

The presence and value of a fingerprint should not be easily spotted by a third party, otherwise it would be potentially removed. In fact, secrecy of our fingerprints is another advantageous property, because our fingerprint encoder, different from image steganography or watermarking, does not directly retouch generated images. As a result, traditional secrecy attack protocols, e.g. Artificial Training Sets (ATS) attack used in (Lerch-Hostalot & Megías, 2016; Yu et al., 2021), is **not** applicable.

Instead, we employ the shadow-model-based attack (Salem et al., 2020) to try detecting the presence and value of a fingerprint from generated images. We assume the attacker can access the model inventor's training data, fingerprint space, and training mechanism. He re-trains his own shadow fingerprint auto-encoder. For the **fingerprint presence attack**, on CelebA dataset, the attacker trains a ResNet-18-based (He et al., 2016) binary classifier to distinguish 10k non-fingerprinted images (5k real plus 5k generated) against 10k generated images from his fingerprinted generators. We find near-saturated 0.981 training accuracy. Then he applies the classifier to 1k inventor's generated images. As a result, we find only **0.505** testing accuracy on the presence of fingerprints, close to random guess. For the **fingerprint value attack**, on CelebA dataset, the attacker applies his shadow decoder (0.991 training bitwise accuracy) to 1k inventor's generated images. As a result, we find only **0.513** testing bitwise accuracy, also close to random guess. We conclude that the mismatch between different versions of fingerprinting systems disables the attacks, which guarantees its secrecy.

### 4.5 ROBUSTNESS AND IMMUNIZABILITY

Deep fakes in the open end may undergo post-processing environments and result in quality deterioration. Therefore, robustness against image perturbations is equally important to our mechanism. When it does not hold for some perturbations, our immunizability property compensates for it.

Following the protocol in (Yu et al., 2019), we evaluate the robustness against five types of image perturbation: cropping and resizing, blurring with Gaussian kernel, JPEG compression, additive Gaussian noise, and random combination of them. We consider two versions of our model: the original version and the immunized version. An immunized model indicates that during training we augment generated images with the corresponding perturbation in random strengths before feeding them to the fingerprint decoder.

It is worth noting that none of the encoder, decoder, and training data are accessible to the public. Therefore, the robustness against perturbation has to be experimented with the black-box assumption, as protocoled in (Yu et al., 2019). In other words, white-box perturbations such as adversarial image modifications (Goodfellow et al., 2015) and fingerprint overwriting, which requires access to the encoder, decoder, and/or training data, are not applicable in our scenario.

We plot in Figure 5 the comparisons of fingerprint detection accuracy among our original/immunized models and the models of (Yu et al., 2021) w.r.t. the strength of each perturbation. We find:

(1) For all the perturbations, fingerprint detection accuracy drops monotonically as we increase the strength of perturbation. For some perturbations in red plots, i.e., blurring and JPEG compression, accuracy drops slowly in a reasonably large range. We consider accepting accuracy $\geq 75\%$. As a result, the robust working range under blurring is: Gaussian blur kernel size $\sim [0, 7]$; under JPEG compression is: JPEG quality $\sim [80, 100]$. Usually, the images turn not functional with perturbations heavier than this range. We, therefore, validate the robustness against blurring and JPEG compression.

(2) For the other perturbations, although our original model is not robust enough, perturbed augmentation compensates significantly in blue dots. We consider accepting accuracy $\geq 75\%$. As a result, the immunized working range under cropping is: cropping size $\sim [60, 128]$; under Gaussian noise

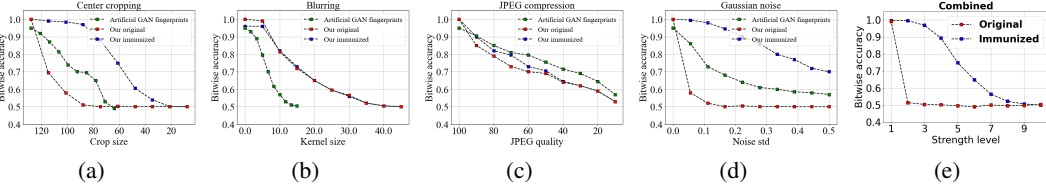

Figure 5: Robustness and immunizability: red/blue plots show, on CelebA, the fingerprint detection of our original/immunized model in bitwise accuracy w.r.t. the strength of perturbations. Green plots show those of (Yu et al., 2021) as references. The data points are directly copied from their paper.

is: noise standard deviation $\sim [0.0, 0.4]$; under combined perturbation is: the combination of the original or immunized working ranges aforementioned. We, therefore, validate the immunizability of our model against cropping, Gaussian noise, and the combined perturbation.

(3) Comparing between (Yu et al., 2021) and our models, for blurring, their model in green plot is less robust than our original/immunized models. For the other perturbations, theirs are more robust than our original models but are outperformed by our immunized models. This indicates the importance of immunizability of a fingerprinting solution, which is however lacking in (Yu et al., 2021).

### 4.6 DEEP FAKE DETECTION AND ATTRIBUTION

The effectiveness, robustness, and immunizability in turn benefit our initial motivation: deep fake detection and attribution. The former task is a binary classification problem to distinguish between real and fake. The latter task is to further finely label the source of a generated image.

We move the solution from *passive* classifiers to *proactive* fingerprinting, and merge the two tasks into one with 1+$N$ classes: 1 real-world source and $N$ GAN sources, where $N$ can be extremely large, as large as our capacity $10^{38}$ in Section 4.2. Then the tasks are converted to verifying if one decoded fingerprint is in our database or not. This is achieved by comparing the decoded fingerprint to each fingerprint in the database given a threshold of bit overlap. According to our $\geq 0.991$ fingerprint detection accuracy, it should be reliable to set the threshold at $128 \times 0.95 \approx 121$. Then the attribution is trivial because we can directly look up the generator instance according to the fingerprint. If the fingerprint is not in the database, it should be a random fingerprint decoded from a real image. We use our immunized model against the combined perturbations in Section 4.5.

**Baselines.** We compare to two state-of-the-art deep fake classifiers (Yu et al., 2019; Wang et al., 2020) as learning-based baselines *passively* relying on inherent visual clues. Because a learning-based method can only enumerate a finite set of training labels, we consider two scenarios for it: closed world and open world. The difference is whether the testing GAN sources are seen during training or not. This does not matter to our method because ours can work with any $N \leq 10^{38}$. For the closed world, we train/evaluate a baseline classifier on 10k/1k images from each of the $N + 1$ sources. For the open world, we train $N + 1$ 1-vs-the-other binary classifiers, and predict as "the other" label if and only if all the classifiers predict negative results. We test on 1k images from each of the real sources or $N$ unseen GAN sources. We in addition refer to (Yu et al., 2021) in comparisons as the other *proactive* but *indirect* model fingerprinting baseline.

**Results.** From Table 3 we find:

(1) Deep fake detection and attribution based on our fingerprints perform equally perfectly ($\sim 100\%$ accuracy) to most of the baselines in the closed world when the number of GAN sources is not too large. However, when $N = 100$, (Yu et al., 2021) is not applicable due to its limited efficiency and scalability. Neither is (Wang et al., 2020) due to its binary classification nature.

| Method | Closed world #GANs | | | Open world #GANs | | |
|---|---|---|---|---|---|---|
| | 1 | 10 | 100 | 1 | 10 | 100 |
| (Yu et al., 2019) | 0.997 | 0.998 | 0.955 | 0.893 | 0.102 | N/A |
| (Wang et al., 2020) | 0.890 | N/A | N/A | 0.883 | N/A | N/A |
| (Yu et al., 2021) | 1.000 | 1.000 | N/A | 1.000 | 1.000 | N/A |
| Ours | 1.000 | 1.000 | 1.000 | 1.000 | 1.000 | 1.000 |

Table 3: Deep fake detection and attribution accuracy on CelebA. A higher value is more desirable. The baseline results are directly copied from (Yu et al., 2021).

(2) Open world is also a trivial scenario to our method but challenges the baseline classifiers (Yu et al., 2019; Wang et al., 2020). When the number of unseen GAN sources increases to 10, (Yu et al., 2019) even degenerates close to random guess. This is a common generalization issue of the learning-based method. (Yu et al., 2021) is still impractical when $N$ is large.

(3) Since deep fake detection and attribution is a trivial task to our method, it makes our advantages independent of the evolution of GAN techniques. It benefits model tracking and pushes forward the emerging direction of model inventors' responsible disclosure.

## 5 CONCLUSION

We achieve responsible disclosure of generative models by a novel fingerprinting mechanism. It allows scalable ad-hoc generation of a large population of models with distinct fingerprints. We further validate its saturated performance in the deep fake detection and attribution tasks. We appeal to the initiatives of our community to maintain responsible release and regulation of generative models. We hope responsible disclosure would serve as one major foundation for AI security.

## REPRODUCIBILITY STATEMENT

The authors strive to make this work reproducible. The appendix contains plenty of implementation details. The source code and well-trained models are available at GitHub.

## ETHICS STATEMENT

This work does not involve any human subject, nor is our dataset related to any privacy concerns. The authors strictly comply with the ICLR Code of Ethics[6].

## ACKNOWLEDGEMENT

Ning Yu was partially supported by Twitch Research Fellowship. Vladislav Skripniuk was partially supported by IMPRS scholarship from Max Planck Institute. This work was also supported, in part, by the US Defense Advanced Research Projects Agency (DARPA) Media Forensics (MediFor) Program under FA87501620191 and Semantic Forensics (SemaFor) Program under HR001120C0124. Any opinions, findings, conclusions, or recommendations expressed in this material are those of the authors and do not necessarily reflect the views of the DARPA. We acknowledge the Maryland Advanced Research Computing Center for providing computing resources. We thank David Jacobs, Matthias Zwicker, Abhinav Shrivastava, and Yaser Yacoob for constructive advice in general.

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

| Layer | CelebA | | LSUN Bedroom | |
|---|---|---|---|---|
| | Bit acc ⇑ | FID ⇓ | Bit acc ⇑ | FID ⇓ |
| 4×4 | 0.953 | 12.06 | 0.693 | 21.44 |
| 8×8 | 0.981 | 12.06 | 0.950 | 21.15 |
| 16×16 | 0.993 | 11.90 | 0.935 | 20.98 |
| 32×32 | 0.991 | 11.07 | 0.894 | 20.24 |
| 64×64 | 0.972 | 10.77 | 0.816 | 19.85 |
| 128×128 | 0.946 | 10.67 | 0.805 | 19.67 |
| Ours (all layers) | 0.991 | 11.50 | 0.993 | 20.50 |

Table 4: Fingerprint detection in bitwise accuracy and generation fidelity in FID w.r.t. the layer to modulate fingerprints. ⇑/⇓ indicates a higher/lower value is more desirable.

# A APPENDIX

## A.1 IMPLEMENTATION DETAILS

We set the length of latent code $d_z = 512$. We non-trivially select the number of bits for fingerprint $d_c = 128$ according to the analysis study on the capacity in Section 4.2. Our encoder $E$ is composed of 8 fully-connected neural layers followed by LeakyReLU nonlinearity, the same as the $\mathcal{Z} \mapsto \mathcal{W}$ mapping network in StyleGAN2 (Karras et al., 2020). The generator $G$ is almost the same as that in StyleGAN2 except we input the latent code $z$ through the input end (replacing the learnable constant tensor) rather than through the modulation. The discriminator $D$ is the same as that in StyleGAN2. The decoder $F$ is almost the same as the discriminator except the output size is adapted to the latent code size plus the fingerprint size $d_z + d_c$.

We train our model using Adam optimizer (Kingma & Ba, 2015) for 400 epochs. We use no exponential decay rate ($\beta_1 = 0.0$) for the first-moment estimates, and use the exponential decay rate $\beta_2 = 0.99$ for the second-moment estimates. The learning rate $\eta = 0.002$, the same as that in StyleGAN2. We train on 2 NVIDIA V100 GPUs with 16GB of memory each. Based on the memory available and the training performance, we set the batch size at 32. The training lasts for about 6 days.

Our code is modified from the GitHub repository of StyleGAN2 (Karras et al., 2020) official TensorFlow implementation (config E)[7]. All the environment requirements, dependencies, data preparation, and command-line calling conventions are exactly inherited.

## A.2 ABLATION STUDY ON MODULATION

In an ablation study, we investigate the effectiveness of fingerprint detection and fidelity of generation when modulating fingerprint embeddings to different generator layers (resolutions).

From Table 4 we find:

(1) For effectiveness, the optimal single layer to modulate fingerprints appears in one of the middle layers, specific to datasets: 16×16 for CelebA and 8×8 for LSUN Bedroom. But our all-layer modulation can achieve comparable or better performance. This should be consistent with different datasets because fingerprint detection turns more effective when we encode fingerprints to more parts of the generator.

(2) For fidelity, the side effect of fingerprinting is less significant if modulation happens in the shallower layer. This is because fingerprinting and generation are distinct tasks, and a shallower modulation leads to less crosstalk. However, considering the FID variance is not significant in general, we regard all-layer modulation as a desirable trade-off between effectiveness and fidelity.

## A.3 ABLATION STUDY ON LOSS TERMS

In another ablation study, on CelebA we investigate the contribution of each loss term in Equation 5 towards fingerprint detection and generation fidelity.

From Table 5 we find:

---

[7] https://github.com/NVlabs/stylegan2

| Loss configuration | Bit acc ⇑ | FID ⇓ |
|---|---|---|
| $\lambda_1 \mathcal{L}_{adv}$ | - | 9.37 |
| $\lambda_1 \mathcal{L}_{adv} + \lambda_3 \mathcal{L}_c$ | 0.999 | 12.84 |
| $\lambda_1 \mathcal{L}_{adv} + \lambda_3 \mathcal{L}_c + \lambda_4 \mathcal{L}_{const}$ | 0.988 | 12.24 |
| $\lambda_1 \mathcal{L}_{adv} + \lambda_3 \mathcal{L}_c + \lambda_4 \mathcal{L}_{const} + \lambda_2 \mathcal{L}_z$ | 0.991 | 11.50 |

Table 5: Fingerprint detection in bitwise accuracy and generation fidelity in FID w.r.t. loss configuration on CelebA. ⇑/⇓ indicates a higher/lower value is more desirable.

(1) For effectiveness, comparing between the first and second rows, solely the fingerprint reconstruction loss term $\mathcal{L}_c$ itself is effective enough to enable saturated fingerprint detection performance.

(2) Comparing between the second and third rows, the consistency loss term $\mathcal{L}_{const}$ improves FID due to the explicit disentanglement learning between latent code and fingerprint representations.

(3) Comparing between the third and fourth rows, the latent code reconstruction term $\mathcal{L}_z$ implicitly improves FID due to its benefit to avoid mode collapse (Srivastava et al., 2017). The collaboration between fingerprint reconstruction and latent code reconstruction through the same decoder further facilitates the disentanglement learning.

### A.4 MORE GENERATED SAMPLES WITH FINGERPRINTS

We show in Figure 6 more samples of LSUN Bedroom and LSUN Cat with high fidelity and disentangled controls of latent code and fingerprint code.

### A.5 FINGERPRINT VISUALIZATION

In order to visualize the imperceptibility and effectiveness of our fingerprinting, in Figure 7 Top we generate images given the same latent code and gradually-varying fingerprint codes. The differences among these images are imperceptible, which supports the fidelity of our generation, and the disentangled control between latent code and fingerprint code. In Figure 7 Bottom we demonstrate the residual images w.r.t. the lest-most image. Because the original differences are so imperceptible, we have to magnify the difference values ×5 to visualize the fingerprint patterns. Our detector effectively learns to attribute different generator sources based on such faint but distinguishable patterns.

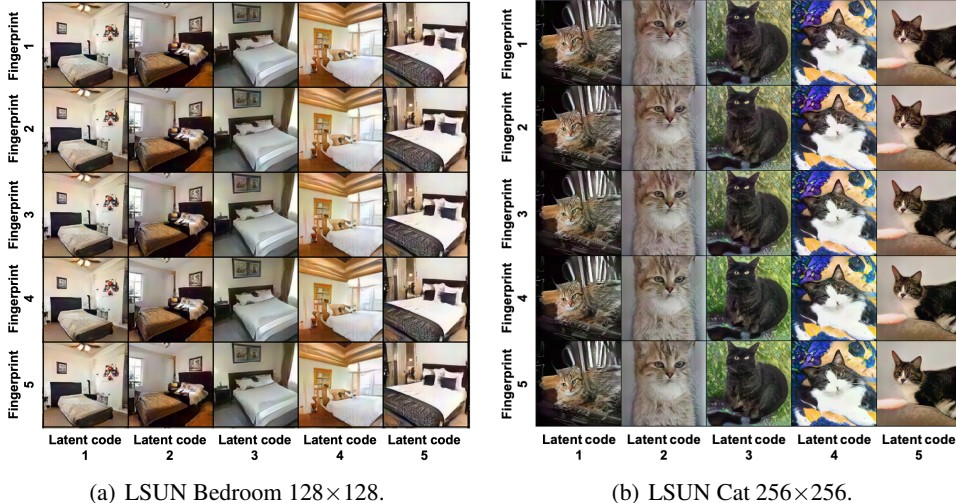

(a) LSUN Bedroom 128×128.        (b) LSUN Cat 256×256.

Figure 6: Fidelity and disentangled control: generated samples from five generator instances. For each row, we use a unique fingerprint to instantiate a generator. For each column, we feed in the same latent code to the generator instances.

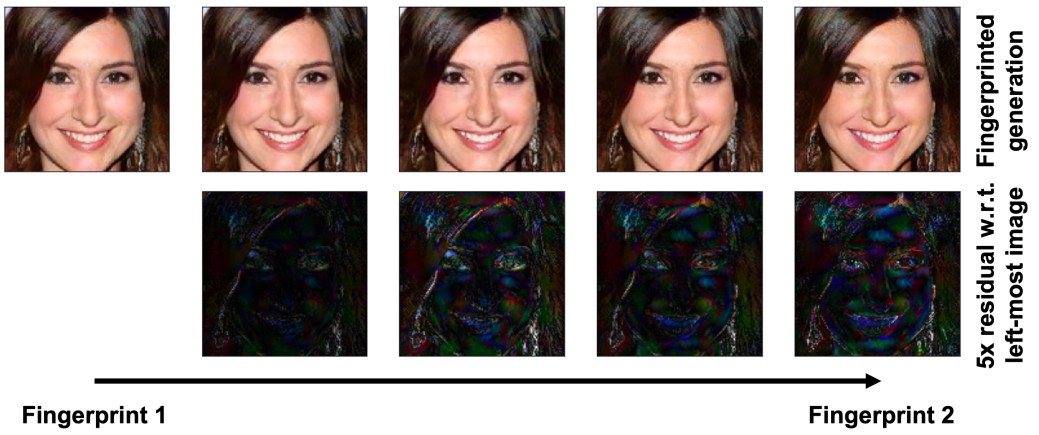

Figure 7: Visualization for the imperceptible patterns of fingerprints. Top: generated images given the same latent code and gradually-varying fingerprint codes from "fingerprint 1" to "fingerprint 2". Bottom: residual images w.r.t. the lest-most image. Because the differences are too faint, we magnify the pixel values $\times 5$ for clear visualization of the fingerprint patterns.

| Method | CelebA | | | LSUN Bedroom | | |
|---|---|---|---|---|---|---|
| | Bit acc $\Uparrow$ | $p$-value $\Downarrow$ | FID $\Downarrow$ | Bit acc $\Uparrow$ | $p$-value $\Downarrow$ | FID $\Downarrow$ |
| SNGAN | - | - | 15.88 | - | - | 26.76 |
| Ours | 0.991 | $< 10^{-36}$ | 15.88 | 0.986 | $< 10^{-34}$ | 28.76 |
| WGAN-GP | - | - | 12.16 | - | - | 21.13 |
| Ours | 0.989 | $< 10^{-36}$ | 13.07 | 0.964 | $< 10^{-31}$ | 22.89 |

Table 6: Fingerprint detection in bitwise accuracy with $p$-value to accept the null hypothesis test, and generation fidelity in FID. $\Uparrow/\Downarrow$ indicates a higher/lower value is more desirable.

## A.6 DIFFERENT GENERIC GAN MODELS

Our fingerprinting design is agnostic to generic GAN models. We replace the StyleGAN2 backbone with SNGAN (Miyato et al., 2018) or WGAN-GP (Gulrajani et al., 2017) backbone, and report our fingerprint bitwise detection accuracy and fidelity on CelebA and LSUN Bedroom datasets.

From Table 6 we find:

(1) Our fingerprinting is generalizable to varying GAN models with near-perfect detection accuracy with $p$-value close to zero, regardless of the original performance of the models.

(2) Our fingerprinting results in negligible $\leq 2.01$ FID degradation. This is a worthy trade-off to introduce the fingerprinting function. In particular for SNGAN on CelebA, our solution does not hurt the original generation quality at all.

