# OpenReview forum: "Responsible Disclosure of Generative Models Using Scalable Fingerprinting"
_ICLR.cc/2022/Conference — ICLR 2022 Spotlight_

### Official Review · Reviewer_yqaN · 2021-11-01

**Correctness:** 4
**Technical Novelty And Significance:** 4
**Empirical Novelty And Significance:** 3
**Recommendation:** 8
**Confidence:** 4

**Main Review:**

# Summary
In this paper, the authors address the problem of fingerprinting GAN (StyleGAN2 in the proposed setting) in a scalable way, e.g. after initial processing, arbitrary fingerprints can be added. In doing so, the authors proposed to leverage modulated convolution layers which scales the output of GAN’s intermedia layers by projection embedded in a channel wise fashion. The authors also provide a set of metrics to evaluate GAN fingerprinting techniques, and accordingly demonstrate the superiority of the proposed method.

# Strong Points

- This paper is well motivated to address the important problem of fingerprinting generative models that are becoming increasingly powerful recently such that provdenting, or at least identifying, its ill-use.
- Overall, this paper is well-structured and easy to follow.
- This paper presents a scalable way of fingerprinting GAN, where after one pass of training, a lot of fingerprints can be added requiring negligible additional resources.
- This paper presents a set of metrics for evaluating the tasks of GAN fingerprinting, which is important as such tasks are still new. Using these metrics, experiments show the advantage of the proposed method.

# Weak Points
- There is no visualization showing the effect of channel-wise modulation as a fingerprinting technique.. It would be helpful for readers to understand such channel-wise effects using visualization, for example like that in [1].
- The experiments are only done with resolutions 128x128 and 256x256. As modern GANs can generate much higher resolution (say, 1024x1024) with fine details, knowing the proposed method’s performance in higher resolution can help justify its motivation dealing with modern GANs.
- In Eq (5) there are four coefficients “ to balance the magnitude of each loss term.“ but how  are they chosen and what is the impact of such a choice are not elaborate? It would be helpful to provide analysis to the choice, as the readers may want to apply the proposed technique to other GAN models which likely have different loss terms and thus magnitudes.

# Questions to Authors
Although this work as presented is interesting, there are some questions I have from reading it thoroughly. They are:
- In Page 3, the author claims that “they can barely sustain a long time against the adversarial iterations of GAN techniques.” Is there any concrete example of fingerprinting tech **not** withstanding adversarial iteration of GANs?
- It seems that the loss term $L_z$ in Eq (2) is orthogonal to other loss terms and is less connected to them. What’s the motivation behind using it besides it’s simply a nice to have from other recent works? It is a must-to-have ingredient in the whole pipeline?


# Other Comments
- As $L_z$ and $L_c$ in Eq (2) and Eq (3) respectively are calculated with the same decoder $F$, it seems to imply that they are calculated on different parts of the logits. It’s better to explicitly mention this fact to avoid confusion.

# Assessment
Overall, I think the strong points( the motivation, novel technique, and important set of metrics for such tasks that follow-up works in this direction would find useful.) are important and overweight weak points (there are more like better to know). So I would recommend a clear acceptance, although the authors are enraged to address may questions/concerns to make the draft more comprehensive.


# Ref
[1] Karras et al., 2021: alias-free generative adversarial networks


# Post-rebuttal

I would like to thank the authors for their response, in which most of my issues are solved.


**Summary Of The Paper:**

In this paper, the authors address the problem of fingerprinting GAN (StyleGAN2 in the proposed setting) in a scalable way, e.g. after initial processing, arbitrary fingerprints can be added. In doing so, the authors proposed to leverage modulated convolution layers which scales the output of GAN’s intermedia layers by projection embedded in a channel wise fashion. The authors also provide a set of metrics to evaluate GAN fingerprinting techniques, and accordingly demonstrate the superiority of the proposed method.

**Summary Of The Review:**

Overall, I think the strong points( the motivation, novel technique, and important set of metrics for such tasks that follow-up works in this direction would find useful.) are important and overweight weak points (there are more like better to know). So I would recommend a clear acceptance, although the authors are enraged to address may questions/concerns to make the draft more comprehensive.

---

> ### Author Response · Authors · 2021-11-20
> **Response to Reviewer yqaN**
>
> We thank all the reviewers for their constructive suggestions, which help improve the completeness of our submission. We are encouraged that reviews are positive in the following five levels:
> - The paper is "**well written**" (Reviewer fu3Z) and "**well-structured**" (Reviewer yqaN).
> - The problem we are researching is “**important**” (Reviewer yqaN) and “**impactful**” (Reviewer GMRc).
> - Our contributions are “**significant**” (Reviewer sqeb) and “**well motivated**” (Reviewer yqaN).
> - Our proactive and scalable solution is “**novel**” (Reviewer sqeb, Reviewer yqaN), “**clever**” (Reviewer fu3Z), and “**sound**” (Reviewer GMRc).
> - Our experiments are “**extensive**” (Reviewer fu3Z) and “**carefully designed**” (Reviewer sqeb).
>
> We now address individual concerns of **Reviewer yqaN** below.
>
> **1. [Visualization showing the effect of fingerprinting.]**
> - In the **revised submission** Appendix Section A.5 in **red** and Figure 7, we visualize the faint but distinguishable fingerprint patterns, which effectively make our fingerprinted generation attributable.
>
> **2. [Higher resolution experiments.]**
> - For computationally efficient iterations and comparisons, we validate our solution on images no larger than 256x256 resolution. Higher resolution generation will leave more space for fingerprint encoding and, as a result, will favor our detection.
>
> **3. [Loss weight settings.]**
> - As mentioned in Section 3.1 in the original submission, we set model weights such that the loss terms after weighing (not the weights themselves) have similar magnitudes. They are empirically set to not bias against any loss term, because each loss term contributes to a properties of our solution. The weight settings are not sensitive. A different weight in the same magnitude level results in comparable fingerprint detection accuracy and generation fidelity. This has been added in **red** in the last paragraph of Section 3.1 in the **revised submission**.
> - We also conduct ablation study for varying loss configurations on CelebA. See the table below as well as Appendix Section A.3 in **red** in the **revised submission**. $L_c$ term solely is effective enough to enable saturated fingerprint detection performance, while $L_{const}$ and $L_z$ terms benefit generation fidelity.
> | Loss configuration                                                                  | Bit acc $\Uparrow$ | FID $\Downarrow$ |
> |-------------------------------------------------------------------------------------|:------------------:|:----------------:|
> | $\lambda_1 L_{adv}$                                                                 |          -         |       9.37       |
> | $\lambda_1 L_{adv}$ + $\lambda_3 L_{c}$                                             |        0.999       |       12.84      |
> | $\lambda_1 L_{adv}$ + $\lambda_3 L_{c}$ + $\lambda_4 L_{const}$                     |        0.988       |       12.25      |
> | $\lambda_1 L_{adv}$ + $\lambda_3 L_{c}$ + $\lambda_4 L_{const}$ + $\lambda_2 L_{z}$ |        0.991       |       11.50      |
> - We also apply our solution with the same loss weight settings to varying GAN models. It always achieves near-perfect detection accuracy with $p$-value close to zero, regardless of the original performance of the models. Our fingerprinting results in negligible $\leq2.01$ FID degradation. This is a worthy trade-off to introduce the fingerprinting function. In particular for SNGAN on CelebA, our solution does not hurt the original generation quality at all.
> | Method  |                    |         CelebA         |                  |   |   |                    |      LSUN Bedroom      |                  |
> |---------|:------------------:|:----------------------:|:----------------:|---|---|:------------------:|:----------------------:|:----------------:|
> |         | Bit acc $\Uparrow$ | $p$-value $\Downarrow$ | FID $\Downarrow$ |   |   | Bit acc $\Uparrow$ | $p$-value $\Downarrow$ | FID $\Downarrow$ |
> | SNGAN   |          -         |            -           |       15.88      |   |   |          -         |            -           |       26.76      |
> | Ours    |        0.991       |       $<10^{-36}$      |       15.88      |   |   |        0.986       |       $<10^{-34}$      |       28.76      |
> | WGAN-GP |          -         |            -           |       12.16      |   |   |          -         |            -           |       21.13      |
> | Ours    |        0.989       |       $<10^{-36}$      |       13.07      |   |   |        0.964       |       $<10^{-31}$      |       22.89      |

---

> ### Author Response · Authors · 2021-11-23
> **Continued Response to Reviewer yqaN**
>
> (Continued)
>
> **4. [Any concrete example of fingerprinting tech not withstanding adversarial iteration of GANs?]**
> - For attackers this is not an easy task as long as the inventor’s well-trained decoder is not leaked. This is validated by our secrecy property in Section 4.4 in the original submission that even if an attacker uses exactly the same fingerprinting technique, as long as he cannot access the well-trained decoder parameters, his shadow fingerprint decoder is not effective in identifying the presence of the inventor’s fingerprints. In other words, a shadow fingerprint decoder will not be effective for adversarially improving GANs.
>
> **5. [$L_z$ loss term is orthogonal to the other loss terms. Is it necessary?]**
> - $L_z$ loss term enables latent code reconstruction. According to Srivastava et al. 2017, it benefits generation diversity and mitigates mode collapse, and therefore implicitly benefits generation fidelity. On the other hand, we pursue the disentanglement between latent code and fingerprint code, as formulated in $L_{const}$. Joint reconstruction for both latent code and fingerprint code using the same decoder facilitates the collaboration between the two codes, resulting in an easier disentangled learning process.These are validated by the table above as well as Appendix Section A.3 in **red** in the **revised submission**.
>
> **6. [$L_z$ and $L_c$ are calculated with the same decoder. They are calculated on different parts of the logits. Please explicitly mention this.]**
> - We have mentioned it in Section 3 Paragraph 2 in the original submission that “decoder F mapping an image to the decoded latent code and fingerprint ($\hat{z}$, $\hat{c}$).”

---

### Official Review · Reviewer_GMRc · 2021-11-02

**Correctness:** 4
**Technical Novelty And Significance:** 3
**Empirical Novelty And Significance:** 3
**Recommendation:** 8
**Confidence:** 3

**Main Review:**

This paper addresses an interesting problem, grounded on concrete and impactful issues raised by the emergence of deep fakes. The authors improve on the literature by addressing the main limitation of previous works, namely the scalability of fingerprinting a lot of generated data. The solution presented in the paper is clear, and the soundness of the approach is well grounded on empirical verifications. I particularly liked that everything I was wondering through my first reading of the paper was eventually addressed somewhere later in the text, either in discussions, or through empirical evidence. Overall, and to the best of my knowledge, I have not found any shortcoming in the paper, although I am not an expert in fingerprinting for GANs.
Regarding the related works, a search on Scholar, Scopus, and DBLP seems to indicate that no major reference is missing regarding the core topic of this submission.

Hereafter are some thoughts & suggestions to challenge this submission a bit.

Maybe, I wish the final section (§5 conclusion) would have brought more discussions & perspectives about potential limitations of the current approach, and what could remain to be done. As an example, I guess from the paper that the authors' approach mostly relies on the assumption that only a few (legitimate) entities are able to build and train state-of-the-art GANs. I may be somewhat speculative due to my lack of expertise in GANs, but I guess that as long as the cost for building such models will decrease, more malicious entities will be able to build their models, thereby mitigating the impact of fingerprinting. It might be valuable to add a short comment in this sense, even if this does not question the good quality of this submission.

Likewise, the model inventor is assumed to be honest here. This naturally goes beyond the scope of this paper, but maybe discussing further works about non-repudiation could represent a nice open question towards the responsible release of industry-wide models.

A few minor remarks can be found hereafter.

**Sec 3.1**
* The parameters $\lambda_i$ are arbitrarily fixed to similar values in order of magnitude, which is not discussed. While I guess that this is beyond the scope of this submission, is there any reason why all the $\lambda_i$ coefficient of the final objective loss should have the same order of magnitude?

**Sec 4.1**
* "Our method results in negligibly < 2.93 FID degrading. This is a reasonable trade-off between effectiveness and fidelity." -> could you elaborate a bit more ? More precisely, are there any reference values for FID with which the non-advised reader may compare ?

**Sec 4.2**
* "on one hand" -> "on the one hand"
* $2^{128 \times 0.991} \approx 10^{38}$ instead of 36 (even better).

**Sec 4.4**
* The description of experiments for fingerprint presence/value attacks are clear, but remain very high-level. I have not found any detail about the implementation and training of the CNN binary classifier, nor any raw performance data. Accordingly, it does not enable us to assess whether enough efforts have been done to reach better accuracy, which may invalidate the secrecy claims. Could you elaborate a bit more on this ?

## Addendum: major overlaps between the Related Works sections of this submission and (Yu et al., 2021)
Since this submission compares to (Yu et al., 2021) a lot, I quickly read this reference and I have found some major overlaps in the related work section (up to minor updates). To the best of my knowledge, this only concerns the related works section. Nevertheless, this may still technically represent some plagiarism. Hence, **I urge the authors to rewrite this section, or at the very least to mention that the related works section is mostly taken from (Yu et al., 2021)**.


**Summary Of The Paper:**

This paper proposes a technique to allow inventors of generated data (e.g. deepfakes) to mark any generated data that may be broadcast through a public API (e.g. like OpenAI did), so that the inventor can easily assess whether a data has been generated through its generative model, and therefore mitigate harmful effects of such fake data. The core idea resides in ploughing a unique fingerprint into the generative model, making it somehow unique for each generated data. The generative model is jointly trained with a decoder whose aim is, given a potentially fake data, to recognize which fingerprint has been used to generate the data. The authors argue that their technique does not cause too much computational overhead, occurs negligible performance degradation compared to standard generative models, and is scalable and robust against potential attacks aiming at circumventing fingerprinting. The claims are verified on several public, GANs-oriented, datasets.

**Summary Of The Review:**

This paper addresses an interesting problem about fingerprinting data generated by GANs, in order to be able to trace any misuse of deepfake. The paper improves state of the art by addressing a major shortcoming of previous works. Overall, the presented solution represents a clear impact on industry-deployed GANs.

---

> ### Author Response · Authors · 2021-11-20
> **Response to Reviewer GMRc**
>
> We thank all the reviewers for their constructive suggestions, which help improve the completeness of our submission. We are encouraged that reviews are positive in the following five levels:
> - The paper is "**well written**" (Reviewer fu3Z) and "**well-structured**" (Reviewer yqaN).
> - The problem we are researching is “**important**” (Reviewer yqaN) and “**impactful**” (Reviewer GMRc).
> - Our contributions are “**significant**” (Reviewer sqeb) and “**well motivated**” (Reviewer yqaN).
> - Our proactive and scalable solution is “**novel**” (Reviewer sqeb, Reviewer yqaN), “**clever**” (Reviewer fu3Z), and “**sound**” (Reviewer GMRc).
> - Our experiments are “**extensive**” (Reviewer fu3Z) and “**carefully designed**” (Reviewer sqeb).
>
> We now address individual concerns of **Reviewer GMRc** below.
>
> **1. [Discussion on the limitation.]**
> - It is true that our solution assumes model inventors to be honest and responsible. But we think this is not a demanding assumption. In fact, the well-acknowledged model inventors in our community, e.g. OpenAI, Deepmind, NVIDIA, are behaving responsibly to earn trust from the society. Our solution turns out to be their favorable option to disclose their responsibility when releasing a powerful generation technique, so as to mitigate the social stress about its unintended misuses by their users. See Page 2 Paragraph 2 in the original submission.
> - It is also true that our solution introduces extra computation loads compared to the original GAN training. Yet this burden belongs to the one-time offline budget, and should be easily accommodated by model inventors, considering those powerful generator techniques that are vulnerable to be misused are always already supported by sufficient computing resources.
> - We agree with the reviewer to add a brief discussion in **red** in the final paragraph of the **revised submission**.
>
> **2. [Loss weight settings.]**
> - As mentioned in Section 3.1 in the original submission, we set model weights such that the loss terms after weighing (not the weights themselves) have similar magnitudes. They are empirically set to not bias against any loss term, because each loss term contributes to a properties of our solution. The weight settings are not sensitive. A different weight in the same magnitude level results in comparable fingerprint detection accuracy and generation fidelity. This has been added in **red** in the last paragraph of Section 3.1 in the **revised submission**.
> - We also conduct ablation study for varying loss configurations on CelebA. See the table below as well as Appendix Section A.3 in **red** in the **revised submission**. $L_c$ term solely is effective enough to enable saturated fingerprint detection performance, while $L_{const}$ and $L_z$ terms benefit generation fidelity.
> | Loss configuration                                                                  | Bit acc $\Uparrow$ | FID $\Downarrow$ |
> |-------------------------------------------------------------------------------------|:------------------:|:----------------:|
> | $\lambda_1 L_{adv}$                                                                 |          -         |       9.37       |
> | $\lambda_1 L_{adv}$ + $\lambda_3 L_{c}$                                             |        0.999       |       12.84      |
> | $\lambda_1 L_{adv}$ + $\lambda_3 L_{c}$ + $\lambda_4 L_{const}$                     |        0.988       |       12.25      |
> | $\lambda_1 L_{adv}$ + $\lambda_3 L_{c}$ + $\lambda_4 L_{const}$ + $\lambda_2 L_{z}$ |        0.991       |       11.50      |
>
> **3. [Clarification for “Negligibly < 2.93 FID degrading”.]**
> - Comparing Row 1 (original StyleGAN2) and Row 5 (ours) in Table 1, we conclude the number 2.93 from the LSUN Cat data. The original FID is 31.01 and our FID is 33.94. The FID difference is 2.93, which is the largest gap over different datasets. But still, this FID difference is small enough and can barely be perceived. One reasonable reference: the best FID number (the highest generation fidelity) that can be achieved by our community is 2.70 (Table 5 Row 4 Column 3 in StyleGAN3 paper), which indicates that the difference in our paper is as imperceptible as the difference between the state-of-the-art generation and its real ground truth.
>
> **4. [Typo of $10^{36}$.]**
> - Revised.

---

> ### Author Response · Authors · 2021-11-20
> **Continued Response to Reviewer GMRc**
>
> (Continued)
>
> **5. [More details about the fingerprint presence attack.]**
> - The attacker’s GAN fingerprinting training technique is exactly the same as the model inventor’s in Section 3. The only difference is that the attacker trains his own version (therefore called shadow model) because he cannot access the inventor’s well-trained version. Then for CelebA dataset, the attacker collects 10k fingerprinted generated images as positive samples, and 5k non-fingerprinted generated images from the original StyleGAN2 plus 5k non-fingerprinted real images as negative samples. He trains a ResNet18-based CNN classifier to distinguish between the two classes and applies it to the inventor’s fingerprinted generated images. We observed near-saturated training performance (0.981 classification accuracy) the random-guess-like testing performance (0.505 classification accuracy) of his classifier, as a result, validating the secrecy of our fingerprinting solution against the fingerprint presence attack. We have added these details in **red** in Section 4.4 in the **revised submission**.
>
>
> **6. [Overlaps in Related Work with Yu et al. 2021.]**
> - We apologize for the concerns! Although using distinct methodologies, our submission and Yu et al. 2021 investigate the same topic. This results in the inevitable overlaps with their Related Work sections. While we were aware of and compliant with the Code of Ethics and had made efforts to reduce the overlaps, we did overlook a small portion of them in the current version (57 overlapped words according to iThenticate). This counts for 0.77% of the entire manuscript. On the way to comply with the suggestions of Reviewer RiQf, we have reduced the overlaps in Section 2 in **red** in the **revised submission**.

---

### Official Review · Reviewer_RiQf · 2021-11-02

**Correctness:** 2
**Technical Novelty And Significance:** 2
**Empirical Novelty And Significance:** 2
**Recommendation:** 6
**Confidence:** 4

**Main Review:**

The idea to actively protect generated synthetic data for bad use is certainly interesting, however I believe this paper cannot be published in its present form since in my opinion there are several issues as described below.

Contribution. The concept of fingerprinting generative models is not new. This has been already recently proposed at ICCV (Yu et al. 2021). For my understanding the main concept is the same, i.e. embedd a fingerprint in the model and enable deepfake detection and attribution by decoding the fingerprints from the generated images. Also note that in Yu et al. authors talk about a proactive and responsible disclosure of pre-trained GAN models. In this respect I find that the first contribution claimed in this submission is not novel. The main difference for my understanding is that the method proposed in Yu et al. cannot scale up to a large number of fingerprints differently from this work. Scalability is hence the main contribution with respect to state-of-the-art, given that performance are very much similar to those obtained by Yu et al. (see experimental results). I think this is not sufficient for a publication at ICLR. It is also worth noting that using an autoencoder-based method to encode a fingerprint into a classifier has been already done in the literature.

Related work. The section about related work is confused in my opinion. In the forensics field watermarking and steganography are two research topics with different goals. Even if they both aim at hiding information, the way this is conducted is fundamentally different. In this regard, I would not mix them in a single paragraph. I believe what is done in this work is more similar to watermarking and the innovation with respect to current literature should be better highlighted. For what concerns network watermarking it is not true that a solution for generative models is missing, see the recent work (Ong et al., CVPR 2021).

Experimental analysis

- Experiments are carried out only on StyleGAN2 model. What happens for different GAN architectures? No generalization analysis is conducted.

- The proposed approach can identify more than 10^36 generator instances, but are these perfectly distinguishable from istances of other generators? This is not clear and should be demonstrated.

- Robustness study is also not sufficient. It is not clear for example if adding adversarial noise can destroy important traces of the fingerprint. More specifically, the ability of the attacker to break the fingerprint has not been studied thoroughly (see Section 4.4.)

- I am very much confused by the experiments carried out in Section 4.6. First, I cannot understand why comparing with passive methods (Wang et al., 2020). Then, for attribution based methods, only one approach has been tested (Yu et al. 2019) while neglecting many others (see references below). Finally, why not comparing with watermarking based methods? It would be very important to show that the proposed method can achieve much better performance than other active approaches.
In this same section authors talk about N GAN sources, I imagine this means the same architecture but a different trained model. This should be made clear, because it can create confusion.

- There are four loss terms in eq.5. I cannot find an ablation study in the experimental section that shows the importance of such terms.

- This work aims at solving the problem of deepfake misuse. Now, I understand that it is possible to perform attribution for a known GAN model. However it is not clear what happens if the synthetic image has not been fingerprinted. Suppose for example that the bad actor builds its own generator without a fingerprint. Will the method confuse the image with a real one? No experiments have been presented to understand the behaviour in this scenario.

=========== Post rebuttal comments ===========

I want to thank the authors for answering to all my comments. Many important points have been clarified
and the paper has been updated accordingly. In particular, I appreciate the new experiments on other GAN architectures
and the ablation study on different loss term configurations. I also find that now the connections with state-of-the-art are much more clear.
In addition, the main contribution as well as the scenario of interest have been better explained. For these reasons I increase my score to borderline accept.

References

- Ong et al. Protecting Intellectual Property of Generative Adversarial Networks from Ambiguity Attacks, CVPR 2021
- Albright and McCloskey, Source Generator Attribution via Inversion, CVPR Workshop 2019
- Asnani et al. Reverse Engineering of Generative Models: Inferring Model Hyperparameters from Generated Images, arXiv 2021
- Girish et al. Towards Discovery and Attribution of Open-world GAN Generated Images, CVPR 2021


**Summary Of The Paper:**

Aim of this work is to develop a method to fingerprint GAN models. In this way, images generated from that model can be detected and attributed to a specific GAN model. This would help in a scenario where a malicious actor uses a published GAN model to produce fake images. In fact, the inserted fingerprint would allow to identify the model and hence to establish that the image is synthetic. From a technical point of view this is achieved by adding a 128-bit fingerprint and jointly training an auto-encoder in a GAN framework. Experiments shows that the proposed approach turns out to be an efficient and scalable solution if compared with state-of-the-art.


**Summary Of The Review:**

My main concerns about this work are related to the contribution, which in my opinion is too limited with respect to state-of-the-art, and also to the insufficient experimental validation. I think that more work is needed in terms of experiments to support the claims, in particular about the generalization, uniqueness and robustness of the fingerprints. For these reasons I believe that this submission is not ready to be published.

UPDATES
Authors have better motivated their proposed method with respect to current literature and also added more experiments, that makes the proposal more convincing. I still believe that some more work on the uniqueness and robustness of the fingerprints should be done, however this can be carried out in a future work.

---

> ### Author Response · Authors · 2021-11-20
> **Response to Reviewer RiQf**
>
> Response to Reviewer RiQf
>
> We thank all the reviewers for their constructive suggestions, which help improve the completeness of our submission. We are encouraged that reviews are positive in the following five levels:
> - The paper is "**well written**" (Reviewer fu3Z) and "**well-structured**" (Reviewer yqaN).
> - The problem we are researching is “**important**” (Reviewer yqaN) and “**impactful**” (Reviewer GMRc).
> - Our contributions are “**significant**” (Reviewer sqeb) and “**well motivated**” (Reviewer yqaN).
> - Our proactive and scalable solution is “**novel**” (Reviewer sqeb, Reviewer yqaN), “**clever**” (Reviewer fu3Z), and “**sound**” (Reviewer GMRc).
> - Our experiments are “**extensive**” (Reviewer fu3Z) and “**carefully designed**” (Reviewer sqeb).
>
> We now address individual concerns of **Reviewer RiQf** below.
>
> **1. [The contributions are not sufficient. Fingerprinting generative models is not new.]**
> - This comment differs from all the other reviewers, who agree on our contributions are “significant” (Reviewer sqeb),  “well motivated” (Reviewer yqaN), “novel” (Reviewer sqeb, Reviewer yqaN), “clever” (Reviewer fu3Z), and “sound” (Reviewer GMRc).
> - We have cited and discussed Yu et al. 2021 in Section 2 Paragraph 2 in the original submission, and have highlighted the technical differences. Our completely novel pipeline improves the fingerprinting efficiency and scalability in orders of magnitudes: “During model deployment, we can fingerprint a generator instance in 5 seconds, contrast to (Yu et al. 2021) that has to retrain a generator instance in 3-5 days. This is a 50000× gain of efficiency.” (Section 4.1). This is a non-trivial and significant contribution on its own. But we partially agree with the reviewer so that we have avoided over-selling the conceptual novelty in **red** in the **revised submission** (Section 1 last paragraph and Section 4.6 last paragraph).
>
> **2. [The performance is very similar to Yu et al., 2021.]**
> - We respectfully disagree. Due to the poor scalability of Yu et al. 2021, in Table 3 Row 3 in the original submission, the two “N/A”s indicate that their solution is unable to handle the scenario if the number of GAN instances is >100. In contrast, our solution in Row 4 is consistently effective regardless of instance size. This difference is non-trivial and should be significant enough to differentiate our solution from Yu et al., 2021.
>
> **3. [Using an autoencoder-based method to encode a fingerprint into a classifier has been already done in the literature.]**
> - This is true, but they do not diminish our novelty to be the first solution that fingerprints a generator backbone and scalably instantiates different generator instances. In fact, we have cited and discussed them in the last paragraph of Related Work in the original submission: "For motivations, the existing works target to fingerprint a single model, while we are motivated by the limitation of Ong et al. 2021 and Yu et al. 2021 to scale up the fingerprinting to as many as $10^{38}$ various generator instances within one-time training. For techniques, most existing works embed fingerprints in the input-output behaviors, while our solution gets rid of such trigger input for improved scalability."
>
> **4. [Watermarking and steganography are two research topics with different goals and should not be mixed. This work is more similar to watermarking.]**
> - Agree. In Section 2 Paragraph 3 in the **revised submission**, we have separated the discussions in **red** of steganography and watermarking. In that paragraph in the original submission, we have highlighted our novel part w.r.t. existing watermarking techniques: "We did not retouch individual images. Rather, our solution is the first study to directly modify generator parameters and encode information into the model."
>
> **5. [Ong et al. 2021 is an existing literature of watermarking for generative models.]**
> - We have cited and discussed it in **red** in the last paragraph of Section 2 in the **revised submission**. In particular, we resolve the scalability limitation of Ong et al. 2021 by getting rid of the trigger input-output behaviors.

---

> ### Author Response · Authors · 2021-11-20
> **Continued Response to Reviewer RiQf**
>
> (Continued)
>
> **6. [Other GAN architectures.]**
> - Our fingerprint modulation design is directly inspired by the stylization design in StyleGAN2 architecture, which was the most recent state-of-the-art at the time of our implementation. It is natural to put a substantial focus on this GAN model.
> - But we agree with the reviewer to show our generalization on other GAN models. See the table below and Appendix Section A.6 in **red** in the **revised submission**. Our fingerprinting is generalizable to varying GAN models with near-perfect detection accuracy with $p$-value close to zero, regardless of the original performance of the models. Our fingerprinting results in negligible $\leq2.01$ FID degradation. This is a worthy trade-off to introduce the fingerprinting function. In particular for SNGAN on CelebA, our solution does not hurt the original generation quality at all.
> | Method  |                    |         CelebA         |                  |   |   |                    |      LSUN Bedroom      |                  |
> |---------|:------------------:|:----------------------:|:----------------:|---|---|:------------------:|:----------------------:|:----------------:|
> |         | Bit acc $\Uparrow$ | $p$-value $\Downarrow$ | FID $\Downarrow$ |   |   | Bit acc $\Uparrow$ | $p$-value $\Downarrow$ | FID $\Downarrow$ |
> | SNGAN   |          -         |            -           |       15.88      |   |   |          -         |            -           |       26.76      |
> | Ours    |        0.991       |       $<10^{-36}$      |       15.88      |   |   |        0.986       |       $<10^{-34}$      |       28.76      |
> | WGAN-GP |          -         |            -           |       12.16      |   |   |          -         |            -           |       21.13      |
> | Ours    |        0.989       |       $<10^{-36}$      |       13.07      |   |   |        0.964       |       $<10^{-31}$      |       22.89      |
>
> **7. [Are fingerprints perfectly distinguishable from instances of other generators?]**
> - We respectfully argue that this question is orthogonal to our research. In fact, our motivation and goal stem from finer granularity. When a new deep fake generation technique is invented, multiple users will request downloads and use it with the same generation functionality. Once a misuse happens, it becomes critical to trace the user who made it. Technically, it means our granularity is to distinguish among different generator instances of the same technique, where each instance corresponds to a different user download. See Page 2 Paragraph 2 in the original submission. Therefore, in this work we train one fingerprint auto-encoder with only one generator technique as the backbone. But we agree with the reviewer that the distinguishability across generator techniques would be an interesting future study.
>
> **8. [Robustness against adversarial noise.]**
> - See Section 4.5 Paragraph 3 in the original submission: "It is worth noting that none of the encoder, decoder, and training data are accessible to the public. Therefore, the robustness against perturbation has to be experimented with the black-box assumption, as protocoled in Yu et al. 2019. In other words, white-box perturbations such as adversarial image modifications and fingerprint overwriting, which requires access to the encoder, decoder, and/or training data, are not applicable in our scenario." Therefore, random noise perturbation as shown in Figure 5d in the original submission is more realistic than adversarial noise to the attacker.
> - However, in case the attacker is able to access the inventor’s fingerprint decoder, and apply the adversarial noise technique [1] to perturb testing images towards a dummy fingerprint code, with epsilon (the perturbation amount limit per pixel) as small as 0.03 (pixel range [0.0, 1.0]), the fingerprint detection results can be biased by the dummy fingerprint, and the encoded fingerprint can be destroyed. Therefore, we urge the inventor to keep the fingerprint decoder secret in the backend, as mentioned at the beginning of Section 3 in the original submission.
> - [1] Madry, Aleksander, et al. "Towards deep learning models resistant to adversarial attacks." ICML 2018.
>
> **9. [Why compare to the passive method Wang et al. 2020?]**
> - Wang et al. 2020 is the SOTA of passive solutions for deep fake detection. We meant to validate it is not easy for passive (learning-based) methods to achieve saturated performance in Table 3. This is one of the motivations for us to investigate an alternative paradigm, the proactive solution based on fingerprint verification.

---

> ### Author Response · Authors · 2021-11-20
> **Continued Response to Reviewer RiQf**
>
> (Continued)
>
> **10. [For the attribution task why only compare to Yu et al. 2019?]**
> - There is not only one attribution method in our comparisons in Table 3. In fact, we also compare to Yu et al. 2021 which is more relevant to our method in terms of its proactive nature, and is a more recent SOTA than any other works referred by the reviewer.
> - We have cited and discussed in **red** in Section 2 Paragraph 1 in the **revised submission** the other methods as the reviewer mentioned. However, we politely disagree to compare to many of them. There are two main reasons. First, our method has already achieved saturated performance in each setting, while more baselines would not disqualify our improvements. Second and more importantly, different from Yu et al. 2021 and our method, all the other methods target to attribute among different generator techniques, including differences in the hyper-parameters space of a generator. They are not designed to differentiate instances (deployed for different user downloads) of the same generator technique. Therefore, comparisons to these references are not well-aligned, and are orthogonal to our motivation to trace users’ responsibility of model misuse.
>
> **11. [Why not compare to watermarking-based methods?]**
> - Similar to Yu et al. 2021, this category of techniques, e.g., Ong et al. 2021, suffers from poor efficiency and scalability. They have to re-train a generator instance every time with a different watermark or a different small set of watermarks, which hinders the model inventors to deploy a large population of well-trained models to different user downloads. Because of this limitation, comparisons to only the most recent SOTA Yu et al. 2021 is representative enough, and is more computationally efficient.
>
> **12. [Are the N GAN sources the same architecture but different trained models?]**
> - No. Our pipeline only needs to train one “mother” generator backbone once. During deployment, we sample N different fingerprint codes, and fingerprint the “mother” generator to obtain N generator instances. This is described in Figure 1 in the original submission, where the gray G indicates the “mother” generator, and the colorful Gs indicate different fingerprinted instances. As a result, the N generator sources are in the same architecture, with the same functionality (Figure 3 in the original submission), and fingerprinted from the same trained model. This one-time training is the core reason for the efficiency and scalability of our solution, which makes a significant difference from the most recent SOTA baseline Yu et al. 2021.
>
> **13. [Ablation study on different loss term configurations.]**
> - See the table below on CelebA dataset, as well as Appendix Section A.3 in **red** in the **revised submission**. $L_c$ term solely is effective enough to enable saturated fingerprint detection performance, while $L_{const}$ and $L_z$ terms benefit generation fidelity.
> | Loss configuration                                                                  | Bit acc $\Uparrow$ | FID $\Downarrow$ |
> |-------------------------------------------------------------------------------------|:------------------:|:----------------:|
> | $\lambda_1 L_{adv}$                                                                 |          -         |       9.37       |
> | $\lambda_1 L_{adv}$ + $\lambda_3 L_{c}$                                             |        0.999       |       12.84      |
> | $\lambda_1 L_{adv}$ + $\lambda_3 L_{c}$ + $\lambda_4 L_{const}$                     |        0.988       |       12.25      |
> | $\lambda_1 L_{adv}$ + $\lambda_3 L_{c}$ + $\lambda_4 L_{const}$ + $\lambda_2 L_{z}$ |        0.991       |       11.50      |
>
> **14. [The application is limited: how if the bad actor build its own model without a fingerprint?]**
> - It is true that our solution assumes model inventors to responsibly corporate. But we think this is not a demanding assumption. In fact, the well-acknowledged model inventors in our community, e.g. OpenAI, Deepmind, NVIDIA, are behaving responsibly to earn trust from the society. Our solution turns out to be their favorable option to disclose their responsibility when releasing a powerful generation technique, so as to mitigate the social stress about its unintended misuses by their users. See Page 2 Paragraph 2 in the original submission. We have also added a brief discussion in **red** in the final paragraph of the **revised submission**. To respond to the reviewer's concern, we appeal to the initiatives of our community to maintain responsible release and regulation of generative models. We hope responsible disclosure would serve as one major foundation for AI security.
> - We appeal to the reviewer to unbias the judgment on this point, by considering **Reviewer GMRc**'s comment: “this does not question the good quality of this submission.”

---

### Official Review · Reviewer_sqeb · 2021-11-02

**Correctness:** 3
**Technical Novelty And Significance:** 4
**Empirical Novelty And Significance:** 3
**Recommendation:** 8
**Confidence:** 4

**Main Review:**

Strengths
1.	This work is significant because it can provide a potential way to detect the deep fake images and attribute the responsibility to a particular user who generated the fake images. Theoretically, this model only needs to be trained once, and a large population of models can be instantiated, each of which has a generator modulated using a different fingerprint.  By doing so, this model has the potential to improve scalability than existing methods that train an auto-encoder to fingerprint the images in the training set and train a GAN for each fingerprint.
2.	This work has technical novelty in that it defines a new loss function for StyleGAN2. This loss function contains a loss term that corresponds to the correctness of the decoded fingerprint, and a loss term corresponds to the consistency of the images generated using the same latent code but different fingerprints. The technical novelty of this work also lies in that this work changed the architecture of the StyleGAN2 backbone by modulating the convolutional filters in the generator with the fingerprint embedding.
3.	This work evaluated the performance of the proposed model using three different datasets. The results are compared with a baseline from the existing work (Yu et al., 2021) that relies on fingerprinting the images in the training data. The experiments are carefully designed, and the results showed that the proposed method could provide the same level of fingerprint detection accuracy and maintain the same level of quality in the generated images as the model from Yu et al., 2021.

Weaknesses
1.	The proposed model is tied to the StyleGAN2 model, while the baseline method (Yu et al. 2021) is agnostic to models. It would be helpful if the authors could demonstrate how the same mechanism can work with other GAN models;
2.	How the scalability of the model is demonstrated can be improved. As shown in section 4.3, the authors only provided the relationship between the detection accuracy and the training set size. The results suggest that there need to be at least 10k samples in the training set to reach high fingerprint detection accuracy in the testing set. It remains unclear if the baseline method, which fingerprints the images of the training set, also requires the same number of or more samples. It is possible that the baseline method requires a smaller sample size than the proposed method. It is also unclear how the training time of the proposed model is compared to the baseline model. It would be helpful if the authors could provide some information concerning the sample size required for the baseline method and the training time required for the baseline method and the proposed method. Alternatively, the authors can provide a brief explanation on this matter.
3.	Even though the proposed method can allow for initiating a large number of models with different fingerprints efficiently once a model is trained, training the model requires at least 10k samples from the fingerprint space. This indicates that even if the model creator only needs to release a small number of models, the training will still need to be conducted to a large amount of fingerprints. If the authors can provide some additional insights about this issue that would be very helpful.
Other comments:
1.	It would be helpful to add some discussion on comparing the robustness and immunizability of the proposed method and the method proposed in Yu et al. 2021.
2.	It seems like the baseline results are directly obtained from the published manuscript; it would be helpful to make this clear.


**Summary Of The Paper:**

This work aims to provide a method to tackle deep fake by making the samples generated using the generative adversarial networks (GANs) contain a user-specific fingerprint that can be accurately detected and attributed to each user.

This paper built and evaluated a GAN model upon the StyleGAN2 backbone. The generator in the proposed GAN is modulated with different fingerprints on the fly. In other words, after training one model with randomly sampled fingerprints and latent code, a large set of models modulated with different fingerprints can be generated within seconds. The experiments results show that the fingerprint can be accurately detected from the GAN generated images, and the fingerprint has a negligible side effect on the original generation quality. The experiments also demonstrated that the optimal number of unique fingerprints this model can accommodate is over 2^128.


**Summary Of The Review:**

Based on the significance and novelty of this work, the reviewer’s recommendation is “Accept”.

---

> ### Author Response · Authors · 2021-11-20
> **Response to Reviewer sqeb**
>
> We thank all the reviewers for their constructive suggestions, which help improve the completeness of our submission. We are encouraged that reviews are positive in the following five levels:
> - The paper is "**well written**" (Reviewer fu3Z) and "**well-structured**" (Reviewer yqaN).
> - The problem we are researching is “**important**” (Reviewer yqaN) and “**impactful**” (Reviewer GMRc).
> - Our contributions are “**significant**” (Reviewer sqeb) and “**well motivated**” (Reviewer yqaN).
> - Our proactive and scalable solution is “**novel**” (Reviewer sqeb, Reviewer yqaN), “**clever**” (Reviewer fu3Z), and “**sound**” (Reviewer GMRc).
> - Our experiments are “**extensive**” (Reviewer fu3Z) and “**carefully designed**” (Reviewer sqeb).
>
> We now address individual concerns of **Reviewer sqeb** below.
>
> **1. [Other GAN models.]**
> - Our fingerprint modulation design is directly inspired by the stylization design in StyleGAN2 architecture, which was the most recent state-of-the-art at the time of our implementation. It is natural to put a substantial focus on this GAN model.
> - But we agree with the reviewer to show our generalization on other GAN models. See the table below and Appendix Section A.6 in **red** in the **revised submission**. Our fingerprinting is generalizable to varying GAN models with near-perfect detection accuracy with $p$-value close to zero, regardless of the original performance of the models. Our fingerprinting results in negligible $\leq2.01$ FID degradation. This is a worthy trade-off to introduce the fingerprinting function. In particular for SNGAN on CelebA, our solution does not hurt the original generation quality at all.
> | Method  |                    |         CelebA         |                  |   |   |                    |      LSUN Bedroom      |                  |
> |---------|:------------------:|:----------------------:|:----------------:|---|---|:------------------:|:----------------------:|:----------------:|
> |         | Bit acc $\Uparrow$ | $p$-value $\Downarrow$ | FID $\Downarrow$ |   |   | Bit acc $\Uparrow$ | $p$-value $\Downarrow$ | FID $\Downarrow$ |
> | SNGAN   |          -         |            -           |       15.88      |   |   |          -         |            -           |       26.76      |
> | Ours    |        0.991       |       $<10^{-36}$      |       15.88      |   |   |        0.986       |       $<10^{-34}$      |       28.76      |
> | WGAN-GP |          -         |            -           |       12.16      |   |   |          -         |            -           |       21.13      |
> | Ours    |        0.989       |       $<10^{-36}$      |       13.07      |   |   |        0.964       |       $<10^{-31}$      |       22.89      |
>
> **2. [Scalability: training size of the baseline Yu et al. 2021.]**
> - As in Section 4.3 in the original submission: “​​We cannot directly compare to (Yu et al., 2021) because it is impractical (time-consuming) to instantiate a large number of their generators for analysis.” As a workaround, we instead trained our detector using <=1k real samples to simulate the non-scalable nature of the baseline, which leads to non-generalizable results. We have highlighted this in **red** in Section 4.3 Paragraph 1 in the **revised submission**.
>
> **3. [Scalability: training time of the baseline Yu et al. 2021 and the proposed method.]**
> - For model training, it takes the baseline 4 hours for fingerprint auto-encoder training, plus 3-5 days for each single generator training. On the other hand, it takes our method 5 days for the fingerprintable generator backbone training. It is worth noting that this is about the one-time offline budget. Yet, as in Section 4.1 in the original submission: “During model deployment, we can fingerprint a generator instance in 5 seconds, in contrast to (Yu et al., 2021) that has to retrain each generator instance in 3-5 days. This is a 50000× gain of efficiency.”

---

> ### Author Response · Authors · 2021-11-23
> **Continued Response to Reviewer sqeb**
>
> (Continued)
>
> **4. [The proposed training always requires 10k+ fingerprint samples even if the model inventor requires only a small number of generator instances.]**
> - Our training has two objectives: generation fidelity (formulated as the adversarial loss) and fingerprint detection accuracy (formulated as the fingerprint reconstruction loss). During training, empirically, generation fidelity converges much more slowly than fingerprint detection accuracy, which means that we have to train for many iterations (sampling >>10k+ fingerprints) anyway in order to pursue good enough generation fidelity.
> - Since we have to sample >>10k+ fingerprints anyway, randomly sampling independent fingerprints on the fly enables us to cover the entire fingerprint code distribution. Otherwise, during testing, if one detected fingerprint does not match the model inventor’s database, we are not sure whether (a) this is because the generator instance does not belong to the inventor’s technique or (b) this is because our detector does not cover this fingerprint code from the beginning.
> - Randomly sampling fingerprints in the entire space during training is cheap. It is a natural choice with nothing to trade-off.
>
> **5. [Robustness and immunizability: comparison to the baseline Yu et al. 2021.]**
> - We have combined Yu et al. 2021 robustness plots and ours in Figure 5 in the **revised submission**. We have added discussion in the last paragraph of Section 4.5 in the **revised submission**. For blurring, their model in green plot is less robust than our original/immunized models. For the other perturbations, theirs are more robust than our original models but are outperformed by our immunized models.
> - The baseline does not investigate the immunizability property, therefore does not train with perturbed data augmentation. It is beyond our scope to study their immunizability.
>
> **6. [Clarify the credits that the baseline results are copied from their manuscript.]**
> - It has been incorporated in **red** in the captions of Table 1 and 3 in the **revised submission**.

---

> > ### Comment · Reviewer_sqeb · 2021-11-29
> > **Response**
> >
> > I have read the authors response and stand by my recommendation.

---

### Official Review · Reviewer_fu3Z · 2021-11-03

**Correctness:** 2
**Technical Novelty And Significance:** 3
**Empirical Novelty And Significance:** 3
**Recommendation:** 6
**Confidence:** 4

**Main Review:**

Pros:
Proactively fingerprinting a model is a clever idea. The advantage of this method is that the model inventor just needs to perform 1 training phase and can deploy plenty models hard-coding random fingerprint to end users. Extensive experiments are reported justifying the choice of the fingerprint code length and the design of the modulated convolution layer.

The paper is well written and easy to follow.

Cons:
(i) My main concern is that the definition of detection and attribution of synthesised images is somewhat misleading. "Detection" usually means if an image is real or generated from *any* GAN models. "Attribution" refers to predicting which GAN source generates a synthesis image. However, in this paper the authors experiment with StyleGAN2 only and the "GAN sources" are actually from a single StyleGAN2 model perturbed with different fingerprint codes. This makes the comparison with Yu 2019 and Wang 2020 a bit unfair. It also raises several questions of whether the proposed method work on other GAN architectures such as SNGAN, ProGAN, etc, also how the fingerprint code could be tracked if each model is trained this way.

(ii) The application side of the proposed method is rather limited, as it requires the model creator to cooperate, or being responsible, from the beginning. It does not solve the current deepfake problem (strictly speaking, GAN synthesised images are not deepfake which is a different terminology). Nevertheless, the authors carefully state a user case that motivates their approach in the Introduction section. While I do not disagree with the authors, I feel the narrow application is a limitation of their proactive approach.

(iii) The authors use bit-wise accuracy to evaluate the fingerprint detection performance throughout their paper. I think there should be an additional metric that is stricter than bit-wise accuracy: a binary metric that evaluates to True if the exact fingerprint code is predicted. This is to evaluate the reliability of the detection model, since 1 bit difference in the detection can lead to a different fingerprint code (similar to cryptographic hash which does not have metric property).

(iv) (Minor). Fig5-b, can the authors elaborate why the immunized model perform worse than the original model when exposed to blurring perturbation?

**Summary Of The Paper:**

This paper addresses the detection and attribution of synthesised images to its GAN sources. The authors propose a proactive method to inject fingerprint into GAN model's parameters during its training. This is done via a modulated convolution layer to integrate the fingerprint into its filter weight and an additional loss term to predict the fingerprint code from the generator's output image. The authors demonstrated that the model fingerprint can be detected from its generated images with high confidence while retaining the synthesis fidelity. The proposed approach is shown robust against common image perturbation sources if these perturbations are shown during training. It also achieves perfect detection and attribution scores when compared with existing methods.

**Summary Of The Review:**

Overall the paper introduce an interesting approach to address the problem of synthesised image detection/attribution. Although the proposed method certainly has merits and applications, I feel that the paper has slightly different terminology of image detection/attribution versus the literature. At the current state of the paper, I am at borderline but would like to hear the authors to elaborate on my points raised above.

Update: I have read the authors' response and other reviews. I think the main advantage of this paper is a novel strategy to fingerprint a GAN model, while the main drawback is that it works on multiple instances of a single GAN source, instead of multiple GAN sources like the baseline approaches. I strongly encourage the authors to clarify this in their next revision.

I upgrade my decision to borderline accept.

---

> ### Author Response · Authors · 2021-11-20
> **Response to Reviewer fu3Z**
>
> We thank all the reviewers for their constructive suggestions, which help improve the completeness of our submission. We are encouraged that reviews are positive in the following five levels:
> - The paper is "**well written**" (Reviewer fu3Z) and "**well-structured**" (Reviewer yqaN).
> - The problem we are researching is “**important**” (Reviewer yqaN) and “**impactful**” (Reviewer GMRc).
> - Our contributions are “**significant**” (Reviewer sqeb) and “**well motivated**” (Reviewer yqaN).
> - Our proactive and scalable solution is “**novel**” (Reviewer sqeb, Reviewer yqaN), “**clever**” (Reviewer fu3Z), and “**sound**” (Reviewer GMRc).
> - Our experiments are “**extensive**” (Reviewer fu3Z) and “**carefully designed**” (Reviewer sqeb).
>
> We now address individual concerns of **Reviewer fu3Z** below.
>
> **1. [Unfair to compare to the two baselines.]**
> - The reviewer’s understanding of “detection” and “attribution” is correct. However, there are multiple types of granularity to evaluate a deep fake detector/attributor. Each solution has its unique advantage in a certain granularity. The previous solutions show distinguishability in the “GAN source” granularity, while we work in finer granularity to distinguish among different model instances of the same source technique. This stems from the important motivation to regulate generator release and trace individual users’ misuses. See Figure 1 and Page 2 Paragraph 2 in the original submission, which receives consents from **Reviewer GMRc** and **Reviewer yqaN**.
> - In Table 3 we meant to highlight previous solutions that work on coarser granularity are not qualified enough for our finer-granularity motivation. The working range of Wang et al. 2020 is the coarsest. It is not designed to handle the attribution task at all (indicated by “N/A” in Row 2). In contrast, our solution can attribute whatever we fingerprinted. This differentiates our solution from Wang et al. 2020 in terms of functionality.
> - Also, we meant to highlight that Yu et al. 2019 fails to handle the open-world attribution tasks (the last two columns Row 1) due to its supervised and non-generalizable nature. In contrast, our solution formulates the attribution tasks as fingerprint matching with no need for supervised training. As a result, we didn’t intend to generalize over the GAN source granularity. Rather, we generalize over the model instance granularity towards unseen generator instances, i.e., from closed-world to open-world without performance degradation. This advantage is rooted in the proactive characteristic of our solution, one of our core motivations to fingerprint a generator.
>
> **2. [Other GAN architectures.]**
> - Our fingerprint modulation design is directly inspired by the stylization design in StyleGAN2 architecture, which was the most recent state-of-the-art at the time of our implementation. It is natural to put a substantial focus on this GAN model.
> - But we agree with the reviewer to show our generalization on other GAN models. See the table below and Appendix Section A.6 in **red** in the **revised submission**. Our fingerprinting is generalizable to varying GAN models with near-perfect detection accuracy with $p$-value close to zero, regardless of the original performance of the models. Our fingerprinting results in negligible $\leq2.01$ FID degradation. This is a worthy trade-off to introduce the fingerprinting function. In particular for SNGAN on CelebA, our solution does not hurt the original generation quality at all.
> | Method  |                    |         CelebA         |                  |   |   |                    |      LSUN Bedroom      |                  |
> |---------|:------------------:|:----------------------:|:----------------:|---|---|:------------------:|:----------------------:|:----------------:|
> |         | Bit acc $\Uparrow$ | $p$-value $\Downarrow$ | FID $\Downarrow$ |   |   | Bit acc $\Uparrow$ | $p$-value $\Downarrow$ | FID $\Downarrow$ |
> | SNGAN   |          -         |            -           |       15.88      |   |   |          -         |            -           |       26.76      |
> | Ours    |        0.991       |       $<10^{-36}$      |       15.88      |   |   |        0.986       |       $<10^{-34}$      |       28.76      |
> | WGAN-GP |          -         |            -           |       12.16      |   |   |          -         |            -           |       21.13      |
> | Ours    |        0.989       |       $<10^{-36}$      |       13.07      |   |   |        0.964       |       $<10^{-31}$      |       22.89      |

---

> ### Author Response · Authors · 2021-11-20
> **Continued Response to Reviewer fu3Z**
>
> (Continued)
>
> **3. [The application is limited: it requires model inventors to responsibly corporate.]**
> - It is true that our solution assumes model inventors to responsibly corporate. But we think this is not a demanding assumption. In fact, the well-acknowledged model inventors in our community, e.g. OpenAI, Deepmind, NVIDIA, are behaving responsibly to earn trust from the society. Our solution turns out to be their favorable option to disclose their responsibility when releasing a powerful generation technique, so as to mitigate the social stress about its unintended misuses by their users. See Page 2 Paragraph 2 in the original submission. We have also added a brief discussion in **red** in the final paragraph of the **revised submission**. To respond to the reviewer's concern, we appeal to the initiatives of our community to maintain responsible release and regulation of generative models. We hope responsible disclosure would serve as one major foundation for AI security.
>
> **4. [Accuracy of the entire fingerprint matching.]**
> - We respectfully argue that this metric is unnecessary because the strict match of the entire fingerprint is unnecessary. For example, given the current bitwise accuracy 0.991, we can calculate the number of mismatched bits in average as 128x(1-0.991)=2 according to our well-trained decoder, and then avoid deploying a new fingerprint code that has a Manhattan distance <2 to any existing fingerprint in the database. In such a way we avoid possible cross-talk of different fingerprint detection due to our non-perfect detection accuracy. But this doesn’t hurt the large capacity of our valid fingerprint space, considering 2^(128-2) is still large enough (Section 4.2 in the original submission).
>
> **5. [The blurring-immunized model performs worse in Figure 5b.]**
> - This is an empirical plot. With blurring perturbations in testing, the performance difference is neglectable, which indicates our original model is robust to blurring by nature. Without blurring perturbation in testing (the left-most dot), the immunized model performs perceptibly worse because the blurring augmentation during training turns out irrelevant and even distractive to the testing.

---

### Decision · Program_Chairs · 2022-01-20

**Decision:**

Accept (Spotlight)

**Comment:**

The paper proposes and studies a method for the responsible disclosure of a fingerprint along with samples generated by a generative model, which has important applications in identifying "deep fakes". The authors establish both the detectability of their fingerprint-without significant loss of fidelity-as well as the robustness to perturbations. The reviewers found the problem and contributions to be important and significant, well substantiated by an extensive experimental study.